# CONFORMALIZED HIERARCHICAL CALIBRATION FOR UNCERTAINTY-AWARE ADAPTIVE HASHING

**Junyu Luo**[1], **Jinsheng Huang**[1], **Yang Xu**[2], **Lutong Zou**[1], **Xiao Luo**[3†], **Bohan Wu**[1],
**Yifan Wang**[4], **Wei Ju**[1], **Ming Zhang**[1†]

[1] State Key Laboratory for Multimedia Information Processing,
School of Computer Science, PKU-Anker LLM Lab, Peking University
[2] School of Mathematical Sciences, Peking University
[3] Department of Statistics, University of Wisconsin–Madison
[4] University of International Business and Economics

```
luo.junyu@outlook.com, hjs@stu.pku.edu.cn, xuyang1014@pku.edu.cn,
bbf@stu.pku.edu.cn, xiao.luo@wisc.edu, wxtpku@pku.edu.cn,
yifanwang@uibe.edu.cn, juwei@pku.edu.cn, mzhang_cs@pku.edu.cn
```

## ABSTRACT

Unsupervised domain adaptive hashing transfers knowledge from labeled source domains to unlabeled target domains, addressing domain shift challenges in real-world retrieval tasks. Existing methods face two critical limitations: target domain noise severely misleads model training, and indiscriminate domain alignment strategies treat all target samples equally, potentially distorting essential feature structures. We propose an uncertainty-aware adaptive hashing approach that addresses these challenges through a hierarchical conformal calibration framework. At the semantic level, we employ conformal inference to generate confidence prediction sets, replacing single pseudo-labels with set-based predictions whose sizes directly quantify sample reliability for weighted pseudo-label learning and domain alignment. This enables the model to focus on reliable samples while suppressing noise. At the representation level, we predict the stability of individual hash bits, where bit-level confidence guides a robust weighted quantization loss and enables dynamic weighted Hamming distance during retrieval, fundamentally enhancing hash code quality and retrieval robustness. Through this hierarchical calibration mechanism, our method achieves more adaptive and robust cross-domain knowledge transfer. Extensive experiments on multiple benchmark datasets demonstrate significant improvements over existing approaches, validating the effectiveness and superiority of our method.

## 1 INTRODUCTION

Efficient approximate nearest neighbor (ANN) similarity retrieval plays a critical role in recommender systems (Tan et al., 2020), visual search (Pu et al., 2025), and retrieval-augmented generation (RAG) (Zhao et al., 2024). Deep hashing, which replaces floating-point distance computations with bitwise operations, offers significant advantages in both retrieval latency and storage costs, making it a key technology for large-scale retrieval systems (Wang et al., 2017; Luo et al., 2023b; Cui et al., 2024). The capability of deep learning models to generate semantically discriminative hash codes has substantially advanced applications.

However, real-world deployment inevitably encounters domain shift: variations in imaging devices, capture styles, and background distributions cause trained hashing models to exhibit semantic confusion and overconfidence in target domains. To bridge this gap, unsupervised domain adaptive hashing (UDAH) has attracted considerable attention (Wang et al., 2023c; Venkateswara et al., 2017; Wang et al., 2023a; Long et al., 2018a; Huang et al., 2021; Wang et al., 2023b; Huang et al., 2020; He et al., 2019). The objective is to transfer labeled source domain knowledge to unlabeled tar-

---

†Corresponding authors.

get domains. Existing domain adaptive hashing methods typically advance along two pathways: ❶ pseudo-labeling, where models generate supervisory signals for target data based on their own predictions (Lee et al., 2013; Xia et al., 2021b), and ❷ domain alignment, which aims to minimize distributional discrepancies between source and target features or adversarial training (Lee et al., 2019b; Ganin et al., 2016; Zhang et al., 2019; Lu et al., 2023a).

While these methods have achieved commendable progress, their performance is often constrained by a fundamental limitation: **unreliable and heuristic handling of model uncertainty**. Existing approaches suffer from three key issues. ❶ They rely on simple heuristics, such as softmax-based confidence thresholding, to filter high-quality pseudo-labels and guide alignment (Sohn et al., 2020). This approach is inherently risky, as softmax scores are not reliable indicators of correctness, as neural networks are prone to overconfident yet erroneous predictions, especially for out-of-distribution samples (Saito & Saenko, 2021; Li et al., 2021), as shown in Figure 1. ❷ They lack verifiable characterization of model uncertainty, with heuristic methods providing no theoretical guarantees and exhibiting extreme sensitivity to manually-tuned thresholds (Chen et al., 2022). ❸ They treat different aspects of uncertainty as a monolithic concept, conflating semantic-level judgment uncertainty with bit-level representation stability uncertainty without employing targeted strategies.

In this work, we argue that unlocking the next generation of powerful UDAH requires moving beyond fragile heuristics toward a principled, multi-level uncertainty quantification framework. We introduce **Co**nformal Hierarchical Ca**l**ibration **A**daptive Hashing (COLA), a novel paradigm that quantifies and leverages uncertainty from semantic to bit levels. The core innovation of COLA

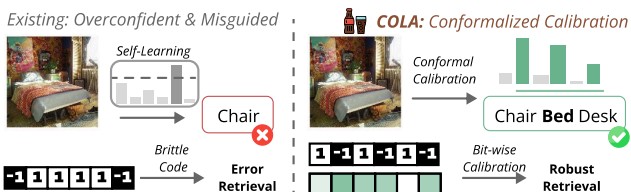

Figure 1: COLA (right) employs hierarchical calibration to replace the single pseudo-labels of existing methods (left) with conformal prediction sets, and weights hash codes with bit-wise confidence to achieve robust retrieval.

lies in its hierarchical conformal calibration framework that provides rigorous statistical guarantees for uncertainty quantification at both semantic and representation levels.

COLA operates through a synergistic two-tier calibration process, as shown in Figure 1. **At the semantic level**, we replace risky point predictions with coverage-controlled prediction sets, whose sizes serve as natural and rigorous measures of semantic uncertainty, enabling more robust pseudo-label learning and domain alignment. **At the representation level**, we introduce a novel bit-level calibration mechanism specifically designed for hashing. We model the reliability of each individual bit in generated hash codes through bit stability prediction, yielding fine-grained hash confidence scores. The score could guide weighted quantization losses during training and, crucially, enable a novel uncertainty-aware weighted Hamming distance during retrieval. Finally, we design a **self-regulating mechanism** that aggregates semantic and bit confidences into endogenous control signals, dynamically balancing pseudo-supervision, domain alignment, and quantization intensity while significantly reducing hyperparameter sensitivity.

Our main contributions can be summarized as follows: ❶ We design a shift from heuristic confidence-based methods to uncertainty quantification frameworks with rigorous statistical guarantees. ❷ We propose COLA that dissects and addresses uncertainty at both semantic and representation levels, yielding more reliable pseudo-supervision and more robust hash codes. ❸ We introduce an elegant self-regulating mechanism that uses quantified uncertainty to dynamically balance multi-objective optimization, achieving truly adaptive learning and enhanced training stability. ❹ Extensive experiments on challenging benchmark datasets demonstrate that COLA significantly outperforms existing state-of-the-art methods.

## 2 RELATED WORK

**Deep Hashing** generating compact binary hash codes to preserve the semantic relationships of data in the Hamming space (Doan et al., 2022; Chen et al., 2024; Tu et al., 2021). This approach significantly reduces storage and computational requirements, making it critical for large-scale retrieval systems (Luo et al., 2023a). Current methods can be fall into two types: supervised (Zhan et al., 2020; Xu et al., 2023; Lu et al., 2023b) and unsupervised (Jin et al., 2020; Wang et al., 2022; Song

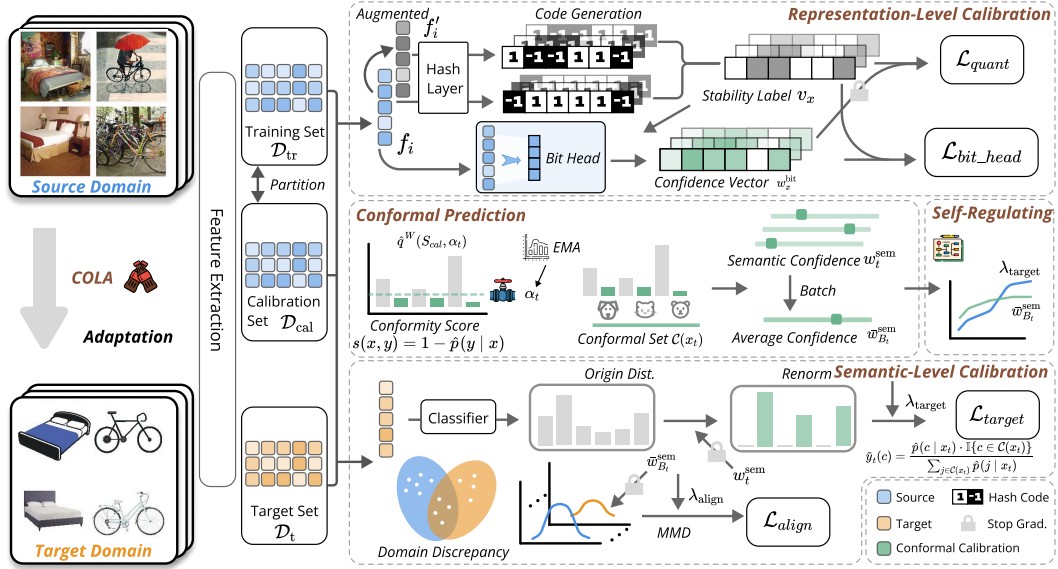

Figure 2: Overview of COLA, a hierarchical conformal calibration framework that addresses uncertainty in domain adaptive hashing through: (1) Semantic-Level Calibration for handling pseudo-label noise (3.2), (2) Representation-Level Calibration for enhancing hash code robustness (3.3), and (3) a Self-Regulating module for dynamically balancing learning objectives (3.4).

et al., 2023; Li et al., 2022; Zhao et al., 2022; Xiao et al., 2023). Unsupervised methods, circumvent the reliance on labels by exploiting the intrinsic structure of the data. However, exitsing methods' retrieval accuracy in practical applications is often affected by potential domain shifts.

**Unsupervised Domain Adaptive Hashing** (UDAH) has emerged as an important research area (Ju et al., 2024; Tang et al., 2024) to address the challenge of domain shift. UDAH targets at transferring knowledge from a label-rich domain to a label-scarce target domain (Long et al., 2018b; He et al., 2022; Lee et al., 2019a). Existing methods typically follow two main strategies: self-learning and domain alignment. Self-learning methods generate supervision for target data based on the model's own prediction (Lee et al., 2013). Domain alignment methods reduce the discrepancy between domains through adversarial training or distribution matching (Huang et al., 2021; Xia et al., 2021a). Despite these advances, existing methods exhibit fundamental limitations in handling model uncertainty, as their strategies are heuristic-driven and inherently unreliable.

**Uncertainty in Retrieval.** Recent works have explored uncertainty modeling in retrieval. (Warburg et al., 2021; 2023) proposed Bayesian metric learning to model aleatoric and epistemic uncertainty via stochastic embeddings. (Tang et al., 2025) utilized probabilistic embeddings for composed image retrieval. In hashing, (Wang & Zhou, 2023; Wang et al., 2025) introduced generative approaches to estimate hash code uncertainty. Unlike these model-based methods, our COLA employs a distribution-free conformal prediction framework. It provides rigorous coverage guarantees under domain shift and hierarchically calibrates uncertainty at both semantic and representation levels without expensive sampling.

# 3 METHODOLOGY

## 3.1 PRELIMINARIES AND OVERVIEW

**Problem Definition.** This work addresses unsupervised domain adaptive hashing (UDAH). Given a labeled source domain $\mathcal{D}_s = \{(x_i^s, y_i^s)\}_{i=1}^{n_s}$ and an unlabeled target domain $\mathcal{D}_t = \{x_j^t\}_{j=1}^{n_t}$ that share the same label space but differ in data distribution, we aim to learn a hash function that maps any input image $x$ to an $L$-bit binary hash code $b \in \{-1, +1\}^L$. The learned function should ensure that semantically similar images have closer distances in Hamming space and enable efficient retrieval.

**Conformal Prediction Basics.** Conformal prediction is a distribution-free framework that constructs prediction sets with rigorous statistical guarantees. Given a calibration set and a user-defined error rate $\alpha$, it produces a set $\mathcal{C}(x)$ for any new $x$ such that the ground truth $y$ is contained in $\mathcal{C}(x)$ with probability at least $1 - \alpha$. This coverage guarantee relies on the exchangeability of data, which we address in the domain adaptation setting via weighted conformal prediction.

**Method Overview.** The core framework of COLA is a hierarchical conformal calibration mechanism that constitutes our primary contribution. COLA consists of two progressive calibration levels:

❶ **Semantic-level calibration** addresses noisy pseudo-labels in the target domain by replacing risky point predictions with theoretically-grounded conformal prediction sets. The size of these sets rigorously quantifies model uncertainty and is directly converted to weights that adaptively suppress the harmful effects of high-uncertainty samples in pseudo-label learning and domain alignment. ❷ **Representation-level calibration** deepens uncertainty analysis to bit-level. We predict the stability of each bit in its generated hash codes. This bit-level confidence guides weighted quantization loss during training and also during retrieval, fundamentally enhancing hash code robustness.

Beyond these core calibration components, we establish a **self-regulating mechanism** as an auxiliary component. This mechanism uses the real-time uncertainty quantified by both calibration levels as intrinsic control signals to adjust the learning focus.

## 3.2 SEMANTIC-LEVEL CONFORMAL CALIBRATION: FROM POINT TO SET

In UDAH, pseudo-label quality fundamentally determines model success. Traditional approaches (Lee et al., 2013; Sohn et al., 2020; Hu et al., 2025) exhibit extreme sensitivity to threshold settings and often fail to ensure pseudo-label reliability under complex domain shifts. To address this fundamental limitation, we introduce conformal prediction theory to establish a semantically uncertain quantification and utilization mechanism with rigorous statistical guarantees. Our core insight abandons high-risk point predictions in favor of constructing prediction sets that theoretically cover the true label with probability $1 - \alpha$ (Liang et al., 2025). The size of this prediction set naturally and rigorously quantifies the predictive uncertainty (encompassing both aleatoric and epistemic uncertainty) for each sample.

### 3.2.1 CONFORMALIZATION VIA CALIBRATION SET

To ensure the calibration set is effective for the target domain, especially under significant domain shifts, we construct it using a targeted selection strategy. First, we extract features for all samples in both the source domain $\mathcal{D}_s$, and the target domain $\mathcal{D}_t$. We then compute the feature centroid of the target domain by averaging all of its feature vectors. Subsequently, for each source sample, we calculate the Euclidean distance between its feature vector and this target centroid. The $r_{\mathrm{cal}}\%$ source samples exhibiting the smallest distances are selected to form the calibration set $\mathcal{D}_{\mathrm{cal}}$. $r_{\mathrm{cal}}$ is set to 20% according to Section 4.2. The remaining source samples constitute the training set $\mathcal{D}_{\mathrm{tr}}$. The targeted selection strategy ensures that our calibration is performed on source data that closely mirrors the characteristics of the target domain, thereby producing more reliable uncertainty estimates for the adaptation task.

We define a conformity score $s(x, y) = 1 - \hat{p}(y \mid x)$ to measure the compatibility between sample $x$ and its true label $y$, where $\hat{p}(y \mid x)$ represents the model's predicted softmax probability. Lower scores indicate stronger model confidence in the prediction.

Subsequently, we compute conformity scores for all samples in $\mathcal{D}_{\mathrm{cal}}$, yielding score collection $\mathcal{S}_{\mathrm{cal}}$. Traditional conformal prediction methods (Vovk et al., 2005; Papadopoulos et al., 2002; Lei et al., 2018) employ a fixed, user-predefined error rate $\alpha$ to calculate quantile threshold $\hat{q}$. However, fixed $\alpha$ fails to adapt to model capability changes throughout the lengthy training process. During early training, overly strict $\alpha$ may result in empty prediction sets, while during later stages, overly lenient $\alpha$ cannot effectively identify uncertainty.

To overcome this limitation, we design a dynamic $\alpha$ adjustment mechanism based on validation accuracy on the source

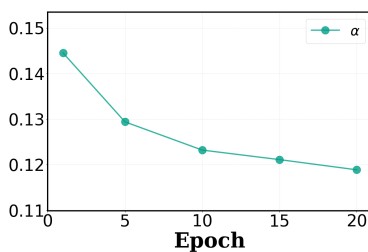

Figure 3: Dynamic $\alpha$ during training on Office-Home Ar $\rightarrow$ Re task.

domain, linearly transforming $\alpha$ from a static hyperparameter into a dynamic variable $\alpha_t$ that evolves with the model's performance. To prevent $\alpha_t$ from fluctuating dramatically due to single evaluation variations, we introduce exponential moving average (EMA) for smooth updates, ensuring adjustment process stability. The dynamic $\alpha$ schedule on Office-Home is illustrated in Figure 3.

According to conformal prediction theory, for a new $x_t$ sampled from the same distribution as the calibration set, the probability that its true label $y_t$ falls within the following prediction set $\mathcal{C}(x_t)$ is at least $1 - \alpha_t$:

$$\mathcal{C}(x_t) = \{y \in \mathcal{Y} \mid s(x_t, y) \leq \hat{q}^{\mathrm{W}}\}, \tag{1}$$

where $\hat{q}^{\mathrm{W}} = \hat{q}^{\mathrm{W}}(\mathcal{S}_{\mathrm{cal}}, \alpha_t)$ denotes the weighted quantile threshold. We sort all conformity scores $\{s(x_i, y_i)\}_{i=1}^{n_{\mathrm{cal}}}$ in ascending order and select the $\lceil (n_{\mathrm{cal}} + 1)(1 - \alpha_t) \rceil$-th value as $\hat{q}^{\mathrm{W}}$.

### 3.2.2 THEORETICAL ANALYSIS

A critical theoretical question arises when applying conformal prediction in UDAH: standard conformal prediction theory requires the calibration data and new test data to be exchangeable, an assumption that breaks down when distribution shifts exist between source and target domains. Therefore, we must address the fundamental question: *do the theoretical coverage guarantees of our constructed prediction sets $\mathcal{C}(x_t)$ remain valid on the target domain?*

To answer this question, building on prior analyses of conformal prediction under covariate or distribution shift (e.g. Barber et al. (2023)), we provide a theoretical analysis demonstrating that our framework remains robust under this limitation. As stated in Theorem 3.1, the coverage guarantee does not completely fail but degrades in a *quantifiable and graceful* manner.

**Theorem 3.1** (Coverage Guarantee under Domain Shift). *Let $d_{\mathrm{TV}}$ denote the total variation distance. Suppose $(X_{\mathrm{train}}, Y_{\mathrm{train}})$ and $(X_{\mathrm{test}}, Y_{\mathrm{test}})$ are random samples from the source and target distributions, respectively. Let $\hat{q}^{\mathrm{W}}$ be derived from equation 1. Then, the following coverage guarantee holds for the target domain:*

$$\mathbb{P}\left(s(X_{\mathrm{test}}, Y_{\mathrm{test}}) \leq \hat{q}^{\mathrm{W}}\right) \geq 1 - \alpha_t - d_{\mathrm{TV}}\left(s(X_{\mathrm{train}}, Y_{\mathrm{train}}), s(X_{\mathrm{test}}, Y_{\mathrm{test}})\right). \tag{2}$$

*If we further assume that the conformal score has a continuous distribution in both domains, then we also have the upper bound:*

$$\mathbb{P}\left(s(X_{\mathrm{test}}, Y_{\mathrm{test}}) \leq \hat{q}^{\mathrm{W}}\right) \leq 1 - \alpha_t + \frac{1}{n + 1} + d_{\mathrm{TV}}\left(s(X_{\mathrm{train}}, Y_{\mathrm{train}}), s(X_{\mathrm{test}}, Y_{\mathrm{test}})\right). \tag{3}$$

The proof and related discussion are provided in Appendix E. This theorem serves as the theoretical foundation of our methodology, revealing the intrinsic logic of synergistic cooperation among various modules in our framework. It implies that our subsequent domain alignment work (detailed in Section 3.2) serves not merely as a heuristic feature distance reduction. By minimizing the latent distribution differences between source and target domains, we are implicitly minimizing the total variation distance between their conformity score distributions.

> **Take Away**: Our theoretical analysis demonstrates that conformal prediction coverage guarantees remain bounded under domain shift. The essential role of domain alignment is to actively reduce the error term in this theoretical bound, making our uncertainty quantification for target domain samples more precise and reliable.

### 3.2.3 UNCERTAINTY-DRIVEN ADAPTIVE LEARNING

With this theoretically-grounded and dynamically-adjustable prediction set $\mathcal{C}(x_t)$, we transform it into effective signals that guide model adaptive learning.

The semantic confidence weight is theoretically grounded in conformal prediction. By Theorem 3.1, smaller prediction sets indicate higher model certainty with statistical coverage guarantees. $|\mathcal{C}(x_t)|$ serves as a natural and rigorous uncertainty measure. We define its reciprocal as the **semantic confidence weight**, which modulates each target sample's contribution:

$$w_t^{\mathrm{sem}} = \frac{1}{|\mathcal{C}(x_t)|}, \qquad \tilde{y}_t(c) = \frac{\hat{p}(c \mid x_t) \cdot \mathbb{I}\{c \in \mathcal{C}(x_t)\}}{\sum_{j \in \mathcal{C}(x_t)} \hat{p}(j \mid x_t)}. \tag{4}$$

This soft label $\tilde{y}_t$ more faithfully reflects the model's judgment within its confidence range compared to hard labels, avoiding overly absolute supervision on uncertain samples.

Combining these two mechanisms, we construct the **semantically-weighted target domain pseudo-supervision loss**. This loss function achieves dual protection: *inter-sample*, through $w_t^{\text{sem}}$ to suppress the overall influence of high-uncertainty samples; *intra-sample*, through $\tilde{y}_t$ to provide smoother and more reliable supervision distribution:

$$\mathcal{L}_{\text{target}} = \frac{1}{|B_t|} \sum_{x_t \in B_t} w_t^{\text{sem}} \cdot \text{CE}\big(\tilde{y}_t, \hat{p}(\cdot \mid x_t)\big). \tag{5}$$

### 3.2.4 CONFIDENCE-GUIDED DOMAIN ALIGNMENT

Beyond constructing more robust single-sample supervision for the target domain, we further apply semantic confidence to guide the macroscopic domain alignment process, correcting the blindness of traditional alignment methods.

Traditional domain alignment approaches uniformly minimize distributional differences between source and target domains. However, when the target domain contains numerous semantically ambiguous *boundary* samples, forced alignment of these samples may actually distort the semantic structure of the shared feature space. To address this issue, we first compute the *average semantic confidence* within a target batch and use it as a weight for the alignment loss:

$$\bar{w}_{B_t}^{\text{sem}} = \frac{1}{|B_t|} \sum_{x_t \in B_t} w_t^{\text{sem}}, \qquad \mathcal{L}_{\text{align}} = \bar{w}_{B_t}^{\text{sem}} \cdot \Big\| \frac{1}{|B_s|} \sum_{x \in B_s} \phi(G(x)) - \frac{1}{|B_t|} \sum_{x \in B_t} \phi(G(x)) \Big\|_2^2. \tag{6}$$

where $G(\cdot)$ denotes the feature extractor and $\phi(\cdot)$ represents the MMD kernel mapping. This batch-level macroscopic weighting mechanism operates under the following logic: when a target batch exhibits low overall confidence, we correspondingly reduce domain alignment intensity to prevent the model from being misled by these *problematic* samples. Conversely, we strengthen alignment when confidence is high. This enables the model to preferentially align core data manifolds with clear semantics in both domains, achieving more stable and meaningful feature distribution alignment that establishes a solid semantic foundation for subsequent high-quality hash code learning.

### 3.3 REPRESENTATION-LEVEL CALIBRATION: BIT-WISE RELIABILITY MODELING

While semantic-level calibration addresses the reliability of *what to learn*, representation-level calibration focuses on the intrinsic stability of *how to learn hash codes effectively*. A high-quality hash code must not only maintain semantic discriminability but also ensure that each individual bit is robust and exhibits low redundancy. The flipping of a single unreliable bit can cause dramatic changes in Hamming distance, severely affecting retrieval precision. Therefore, we extend uncertainty analysis from the macroscopic semantic level to the microscopic bit level.

**Proxy Task for Bit Stability.** To quantify the reliability of each bit, we design a self-supervised proxy task. The core assumption is that a robust bit should maintain sign stability when facing minor perturbations in input data.

Specifically, for each sample $x_i$ in the source domain, we obtain its feature vector $f_i$. An augmented version $f_i'$ is created by applying Gaussian noise to $f_i$. After passing $f_i$ and $f_i'$ through the hash layer, they yield continuous pre-hash vectors $h_i$ and $h_i'$. Based on these, we generate a stability label $v_{i,k}$ for each bit $k$ of $h_i$:

$$v_{i,k} = \mathbb{I}\{\text{sign}(h_{i,k}) = \text{sign}(h_{i,k}')\}, \tag{7}$$

where $\mathbb{I}(\cdot)$ denotes the indicator function. Then, we introduce a lightweight bit confidence prediction head $G_{\text{bit}}(\cdot)$ that operates in parallel with the backbone network. It receives image features and predicts an $L$-dimensional confidence vector $w_{x,k}^{\text{bit}} \in [0,1]^L$. This head is trained through binary cross-entropy loss to predict $v_{x,k}$, which naturally drives the predicted confidence to polarize toward binary values $\{0,1\}$ without requiring explicit thresholds. We employ a separate prediction head rather than on-the-fly perturbation during inference to ensure retrieval efficiency. Direct perturbation would require multiple forward passes per query, significantly increasing latency. Our lightweight head predicts stability in a single pass ($O(1)$), maintaining the speed advantage of hashing.

### 3.3.1 CONFIDENCE-GUIDED HASH LEARNING AND RETRIEVAL

The learned bit confidence $w^{\mathrm{bit}}$ plays a crucial role in both training and testing phases, enabling end-to-end uncertainty awareness.

**Weighted Quantization Loss.** Traditional quantization loss $\|h - \mathrm{sign}(h)\|$ uniformly penalizes all bits that deviate from $\pm 1$. We leverage bit confidence to weight this loss, making the model focus more on bits predicted to be stable and reliable during training, while providing greater tolerance for unstable bits and allowing them more thorough exploration in continuous space.

We reweight the quantization term using bit confidence, applying stronger constraints only on *trustworthy bits* while reducing backward noise from unstable bits:

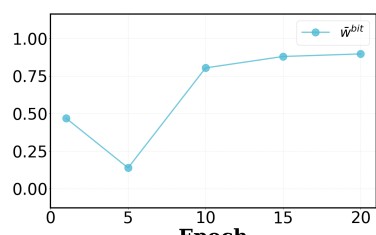

$$\mathcal{L}_{\mathrm{quant}} = \frac{1}{|B|L} \sum_{x \in B} \sum_{k=1}^{L} \mathrm{stop\_grad}(w_{x,k}^{\mathrm{bit}}) \cdot \max\left(0, 1 - |h_{x,k}|\right), \tag{8}$$

where $\mathrm{stop\_grad}$ prevents the model from circumventing quantization by manipulating $w^{\mathrm{bit}}$. The dynamics of $w^{\mathrm{bit}}$ are illustrated in Figure 4, where $\bar{w}^{\mathrm{bit}}$ is the average of $w^{\mathrm{bit}}$ in a mini-batch.

Figure 4: Bit-level calibration $\bar{w}^{\mathrm{bit}}$ on Office-Home Ar→Re task.

**Uncertainty-aware Weighted Hamming Distance.** During retrieval, we leverage the learned bit confidence to dynamically weight Hamming distance, suppressing contributions from unreliable bits. We use the bit confidence $w_q^{\mathrm{bit}}$ of query sample $x_q$ as dynamic weights to construct a novel distance metric. This ensures that when computing distances between query and database samples, bit positions where the query sample itself exhibits uncertainty receive lower weights, naturally reducing the noise impact from unreliable bit flips and significantly enhancing retrieval robustness:

$$d_{\mathrm{UWHD}}(x_q, x_d) = \sum_{k=1}^{L} w_{q,k} \cdot \frac{1}{2}\left(1 - b_{q,k}b_{d,k}\right), \tag{9}$$

where $w_{q,k} \in [0,1]$ represents the bit-level weight derived from query confidence $w_q^{\mathrm{bit}}$. Note that Eq. 9 utilizes continuous weights primarily for differentiable optimization during training. For efficient large-scale retrieval, we binarize the query weights $w_{q,k} \in \{0,1\}$ via rounding. This reduces the metric to a masked Hamming distance, enabling UWHD to be computed via efficient bitwise operations. This series of designs enables our model to not only learn hash code generation but also develop quality assessment capabilities for its own generated hash codes, integrating this assessment ability throughout the entire lifecycle from learning to application.

### 3.4 SELF-REGULATING CALIBRATED ADAPTATION: A CLOSED-LOOP LEARNING SYSTEM

We further construct a self-regulating mechanism that transforms these tools from passive to active components. This mechanism addresses the classic challenge of manually balancing loss weights in multi-objective optimization by using the model's real-time uncertainty as intrinsic control signals to dynamically adjust learning focus, forming an intelligent closed-loop system.

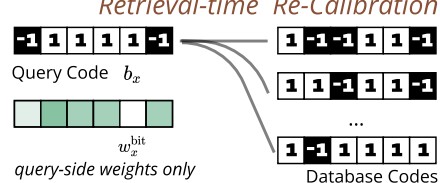

Figure 5: Illustration of bit-level uncertainty calibration mechanism.

Specifically, we compute the average semantic confidence $\bar{w}_{B_t}^{\mathrm{sem}}$ and average bit confidence $\bar{w}_{B_t}^{\mathrm{bit}}$ for each batch $B_t$. These aggregated indicators reflect the model's overall grasp of the current batch data at the present stage. We use them as inputs to adaptive weights $\lambda(\cdot)$ to dynamically modulate the intensity of various loss terms:

$$\lambda_{\mathrm{target}}(B_t) = f_{\mathrm{sem}}(\mathrm{stop\_grad}(\bar{w}_{B_t}^{\mathrm{sem}})), \qquad \lambda_{\mathrm{quant}}(B_t) = f_{\mathrm{quant}}(\mathrm{stop\_grad}(\bar{w}_{B_t}^{\mathrm{bit}})), \tag{10}$$

where $f(\cdot)$ represents linear scaling functions, and the $\mathrm{stop\_grad}$ operation ensures these weights do not directly participate in gradient computation, guaranteeing training stability.

The intuitive logic follows a natural learning progression. During early training, with low $\bar{w}^{\mathrm{sem}}$, resulting in small $\lambda_{\mathrm{target}}$ and $\lambda_{\mathrm{align}}$, the model treats pseudo-labels and domain alignment cautiously,

Table 1: Performance of cross-domain retrieval (mAP%) on Office-Home and Office-31.

| Methods | OFFICE-HOME | | | | | | OFFICE-31 | | | | | | |
|---|---|---|---|---|---|---|---|---|---|---|---|---|---|
| | Pr→Re | Cl→Re | Re→Ar | Re→Pr | Re→Cl | Ar→Re | Am→Ds | Am→We | We→Ds | Ds→Am | We→Am | Ds→We | Avg. |
| ITQ | 26.81 | 14.83 | 25.37 | 28.19 | 14.92 | 25.88 | 29.55 | 28.53 | 58.00 | 26.83 | 25.09 | 58.89 | 30.24 |
| OCH | 18.65 | 10.27 | 17.54 | 20.15 | 10.05 | 18.09 | 24.86 | 22.49 | 51.03 | 22.45 | 20.79 | 53.64 | 24.17 |
| DSH | 8.49 | 5.47 | 9.67 | 8.26 | 5.28 | 9.69 | 16.66 | 15.09 | 39.24 | 16.33 | 13.58 | 41.07 | 15.74 |
| SGH | 24.51 | 13.62 | 22.53 | 25.73 | 13.51 | 22.93 | 24.98 | 22.47 | 53.94 | 22.17 | 20.52 | 56.36 | 26.94 |
| GraphBit | 18.18 | 16.87 | 11.51 | 10.81 | 18.91 | 21.32 | 24.48 | 23.12 | 22.09 | 53.82 | 21.34 | 51.43 | 24.49 |
| GTH-g | 20.00 | 10.99 | 18.28 | 21.95 | 11.68 | 19.05 | 23.08 | 21.20 | 49.38 | 19.52 | 17.41 | 50.14 | 23.56 |
| PWCF | 34.03 | 24.22 | 28.95 | 34.44 | 18.42 | 34.57 | 39.78 | 34.86 | 67.94 | 35.12 | 35.01 | 72.91 | 38.35 |
| DHLing | 48.47 | 30.81 | 38.68 | 45.24 | 25.15 | 43.30 | 41.96 | 45.10 | 75.23 | 42.89 | 41.74 | 79.91 | 46.54 |
| DAPH | 27.20 | 15.29 | 27.35 | 28.19 | 15.29 | 26.37 | 32.80 | 28.66 | 60.71 | 28.66 | 27.59 | 64.11 | 31.85 |
| PEACE | 53.04 | 38.72 | 42.68 | 54.39 | 28.36 | 45.97 | 46.69 | 48.89 | 78.82 | 46.91 | 46.95 | 83.18 | 51.22 |
| DANCE | 53.73 | 39.03 | 43.54 | 55.14 | 28.87 | 44.53 | 44.78 | 47.66 | 78.39 | 46.68 | 48.61 | 84.75 | 51.31 |
| IDEA | 59.18 | 45.71 | 49.64 | 61.84 | 32.77 | 51.19 | 48.70 | 54.43 | 84.97 | 53.53 | 53.71 | 88.69 | 57.03 |
| COUPLE | 63.94 | 49.24 | 54.35 | 64.29 | 41.39 | 54.14 | 50.27 | 59.32 | 85.26 | 56.04 | 56.35 | 88.90 | 60.29 |
| **COLA** | **67.04** | **52.65** | **57.23** | **67.88** | **41.71** | **57.35** | **52.51** | **62.08** | **87.28** | **58.09** | **57.60** | **89.65** | **62.59** |

Table 2: Performance of cross-domain retrieval (mAP%) on MNIST and USPS.

| Methods | MNIST → USPS | | | | | | USPS → MNIST | | | | | | |
|---|---|---|---|---|---|---|---|---|---|---|---|---|---|
| | 16 | 32 | 48 | 64 | 96 | 128 | 16 | 32 | 48 | 64 | 96 | 128 | Avg. |
| ITQ | 13.05 | 15.57 | 18.54 | 20.12 | 23.12 | 23.89 | 13.69 | 17.51 | 20.40 | 20.30 | 22.79 | 24.59 | 19.46 |
| OCH | 13.73 | 17.22 | 19.59 | 20.18 | 20.66 | 23.34 | 15.51 | 17.75 | 18.97 | 21.50 | 21.27 | 23.68 | 19.45 |
| DSH | 20.60 | 22.21 | 23.68 | 24.28 | 25.73 | 26.50 | 19.54 | 21.22 | 22.89 | 23.79 | 25.91 | 26.46 | 23.57 |
| SGH | 14.24 | 16.69 | 18.72 | 19.70 | 21.00 | 21.95 | 13.26 | 17.71 | 18.22 | 19.01 | 21.69 | 22.09 | 18.69 |
| GraphBit | 13.92 | 17.86 | 20.17 | 20.82 | 21.32 | 23.19 | 15.16 | 16.82 | 17.87 | 19.85 | 20.10 | 22.54 | 19.13 |
| GTH-g | 20.45 | 17.64 | 16.60 | 17.25 | 17.26 | 17.06 | 15.17 | 14.07 | 15.02 | 15.01 | 14.80 | 17.34 | 16.47 |
| PWCF | 47.47 | 51.99 | 51.44 | 51.75 | 50.89 | 59.35 | 47.14 | 50.86 | 52.06 | 52.18 | 57.14 | 58.96 | 52.60 |
| DHLing | 49.24 | 54.90 | 56.30 | 58.28 | 58.80 | 59.14 | 50.14 | 51.35 | 53.67 | 58.65 | 58.42 | 59.17 | 55.67 |
| DAPH | 25.13 | 27.10 | 26.10 | 28.51 | 30.53 | 30.70 | 26.60 | 26.43 | 27.27 | 27.99 | 30.19 | 31.40 | 28.16 |
| PEACE | 52.87 | 59.72 | 60.69 | 62.84 | 65.13 | 68.16 | 53.97 | 54.82 | 58.69 | 60.91 | 62.65 | 65.70 | 60.51 |
| DANCE | 53.18 | 57.98 | 61.23 | 63.15 | 65.92 | 68.87 | 54.31 | 55.64 | 57.26 | 61.49 | 63.43 | 66.23 | 60.72 |
| IDEA | 58.89 | 64.48 | 65.72 | 67.48 | 70.24 | 74.34 | 60.99 | 61.47 | 65.45 | 67.97 | 69.72 | 72.31 | 66.59 |
| COUPLE | 60.56 | 66.05 | 66.23 | 67.98 | 73.02 | 75.12 | 63.28 | 64.94 | 67.44 | 70.19 | 72.87 | 74.62 | 68.53 |
| **COLA** | **62.21** | **67.72** | **67.35** | **68.91** | **75.09** | **77.67** | **65.11** | **67.27** | **69.83** | **72.94** | **74.33** | **76.33** | **70.40** |

avoiding aggressive adaptation before sufficiently understanding the target domain. This mechanism naturally acts as a warm-up strategy: early in training, high uncertainty leads to low $\lambda_{\text{target}}$, preventing the model from overfitting to noisy pseudo-labels. As the model learns from the source domain, uncertainty decreases, and the target adaptation gradually engages.

Finally, our total training objective integrates all modules through dynamic balancing via the self-regulating mechanism:

$$\mathcal{L}_{\text{total}} = \mathcal{L}_{\text{source}} + \mathcal{L}_{\text{bit\_head}} + \lambda_{\text{target}}\mathcal{L}_{\text{target}} + \lambda_{\text{align}}\mathcal{L}_{\text{align}} + \lambda_{\text{quant}}\mathcal{L}_{\text{quant}} \tag{11}$$

This design paradigm enables the entire adaptation process to be governed by the model's own cognitive state, achieving truly adaptive, robust, and efficient end-to-end learning.

**Computational Complexity.** The calibration phase requires only one-time sorting and weighted quantile estimation with complexity $O(n_{\text{cal}} \log n_{\text{cal}})$, where $n_{\text{cal}}$ is the size of the calibration set. This can be approximately reduced to linear time using quantile sketching algorithms.

## 4 EXPERIMENT

### 4.1 EXPERIMENTAL SETTINGS

**Datasets.** We compare our COLA with extensive baselines on three popular cross-domain benchmarks: Office-Home (Venkateswara et al., 2017), Office-31 (Saenko et al., 2010), and Digits(MNIST (LeCun et al., 1998) and USPS (Hull, 1994),). We follow the transfer tasks as in previous research (Wang et al., 2023b; Luo et al., 2025) for fair comparison.

**Baselines.** We compare our COLA with state-of-the-art hashing methods, including five unsupervised methods (ITQ (Gong et al., 2012), OCH (Liu et al., 2018), DSH (Jin et al., 2013), SGH (Jiang & Li, 2015), GraphBit (Wang et al., 2022)) and eight domain-adaptive methods (GTH-g (Zhang et al., 2019), PWCF (Huang et al., 2020), DHLing (Xia et al., 2021a), DAPH (Huang et al., 2021), PEACE (Wang et al., 2023a), DANCE (Wang et al., 2023b), IDEA (Wang et al., 2023d), COUPLE (Luo et al., 2025)) as baselines. More details can be found in Appendix C.

Table 3: Comparison of model variants on the Office-Home with 64 bit hash code.

| Variants | SC | RC | SR | Pr→Re | Cl→Re | Re→Ar | Re→Pr | Re→Cl | Ar→Re | Avg. |
|---|---|---|---|---|---|---|---|---|---|---|
| COLA (None) | | | | 59.65 | 46.47 | 49.66 | 59.56 | 31.92 | 47.56 | 49.14 |
| COLA-SC | ✓ | | | 59.92 | 47.08 | 49.89 | 59.81 | 32.07 | 47.82 | 49.43 |
| COLA-RC | | ✓ | | 60.83 | 48.64 | 51.33 | 61.22 | 35.84 | 52.70 | 51.76 |
| COLA-SR | | | ✓ | 61.41 | 47.77 | 52.24 | 61.69 | 35.11 | 51.06 | 51.55 |
| COLA w/o SC | | ✓ | ✓ | 65.13 | 48.89 | 54.12 | 65.97 | 40.01 | 53.04 | 54.53 |
| COLA w/o RC | ✓ | | ✓ | 65.95 | 49.39 | 56.16 | 66.87 | 39.68 | 53.18 | 55.21 |
| COLA w/o SR | ✓ | ✓ | | 65.25 | 49.30 | 53.29 | 65.62 | 39.95 | 55.03 | 54.74 |
| **COLA** (Full Model) | ✓ | ✓ | ✓ | **67.04** | **52.65** | **57.23** | **67.88** | **41.71** | **57.35** | **57.31** |

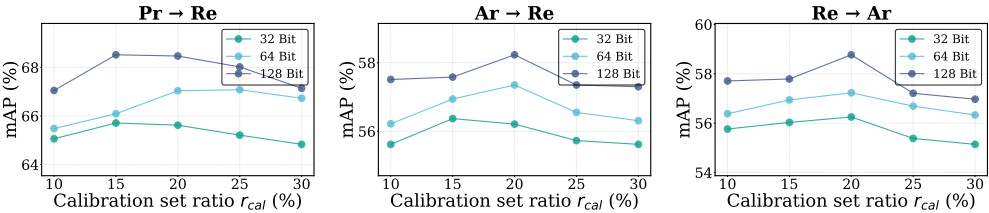

Figure 6: Sensitivity analysis on calibration set ratio $r_{cal}$.

**Implementation Details.** To guarantee a fair comparison, our model configuration is set as in (Wang et al., 2023d; Luo et al., 2025). All experiments are implemented in PyTorch and conducted on a single NVIDIA Hopper GPU. The hash layer consists of a two-layer MLP, and the same structure is used for bit head prediction. We use the Adam optimizer and the initial learning rate is set to 0.001. The batch size is set to 32. The training epoch is set to 35. And we set the proportion of the calibration set to 0.2, the mapping range for $\alpha$ to [0.05, 0.2], and EMA smoothing coefficient to 0.7 as common practice.

**Evaluation Metrics.** We use several standard metrics to assess our COLA: mean Average Precision (mAP), Top-N accuracy curve, Top-N recall curve, and precision-recall curve. The mAP is widely adopted to measure the overall retrieved list. Top-N accuracy and recall curves can demonstrate the performance given varying retrieval quantities. The precision-recall curve is also widely popular due to the trade-off between the precison and recall.

## 4.2 EMPIRICAL RESULTS

**Performance Comparison.** To begin with, we compare the retrieval performance of all approaches on three benchmark datasets, as shown in Tables 1 and 2. Table 1 reports the cross-domain retrieval results on Office-Home and Office-31 with a fixed 64-bit hash code length. Furthermore, we investigated the cross-domain performance of each method under varying hash code lengths on the USPS and MNIST datasets, and the results are presented in Table 2. From the results reported in Tables 1 and 2, we observe that COLA consistently and significantly outperforms existing state-of-the-art approaches, achieving an average improvement of around 3.3% in retrieval performance. Earlier methods, such as DAPH (Huang et al., 2021), PEACE (Wang et al., 2023a) generally suffer from inferior performance due to relatively simplistic domain adaptation strategies. The performance gain of our COLA over advanced baselines can be largely attributed to its unique hierarchical uncertainty calibration framework. To better understand our COLA, we also conduct qualitative analysis experiments. In particular, precision-recall curves, Top-N precision curves, and Top-N recall curves are compared. The more detailed qualitative analysis of these results is provided in the Appendix D.1.

**Ablation Study.** Table 3 reports the retrieval performance of ablation variants on the Office-Home dataset. We can conclude that the COLA is of best performance, demonstrating the importance of each component. The variant *COLA (None)* in Table 3 represents the baseline with all three components (SC, RC, SR) removed, which serves as a standard UDAH baseline. Disabling semantic-level calibration leads to a significant performance drop from 57.31% to 54.53% in average mAP. This underscores the critical role of conformal prediction in quantifying semantic uncertainty to generate reliable soft pseudo-labels and guide domain alignment. Excluding representation-level calibration

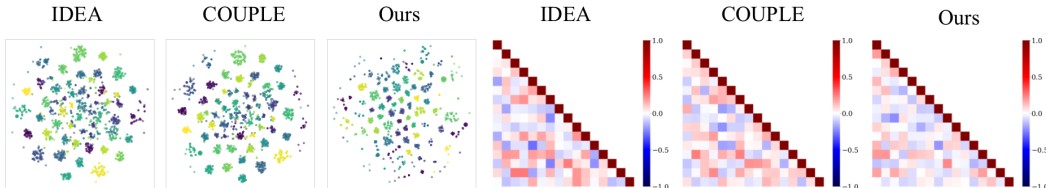

Figure 7: The t-SNE visualization of three methods on Office-Home; The correlation heatmaps of 16-bit binary codes on Office-Home (Ar → Re).

degrades the mAP to 55.21%, which demonstrates the importance of modeling bit-wise reliability. These substantial performance degradations empirically validate the necessity of our weighting mechanisms, with each component contributing meaningfully to the final performance. Also, removing the self-regulating results in a notable performance decrease to 54.74%. This confirms the benefit of dynamically balancing the learning objectives based on the model's real-time uncertainty. We also compared with other variants that contain a single component. Note that the centroid-based calibration set is essential for the SC module to compute semantic distances, and dynamic $\alpha$ is inherent to the SR mechanism. Thus, our ablation design properly isolates the contribution of each component within the coherent framework. More details of the ablation study are in the Appendix D.2.

**Uncertainty Analysis.** We analysed the bit-level confidence of target domain across different hash lengths (32, 64, and 128 bits) on Office-Home dataset. From the results shown in figure 8, we can draw the following conclusions. Firstly, the bit-level confidence starts near 0.5, reflecting the model's random initial state, then sharply drops to near-zero as the model begins learning and calibrating its uncertainty. Finally, as the model converges and learns a stable feature representation, the confidence rises to a high and stable value. Secondly, shorter codes (32-bit) achieve a higher final confidence, as each bit must be more informative. Conversely, longer codes (128-bit) require more time to stabilize, resulting in a slower confidence recovery during training.

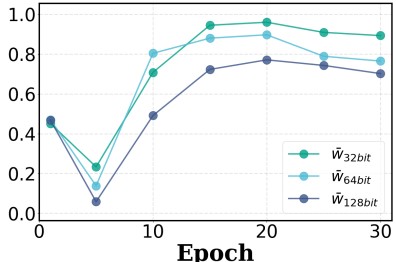

Figure 8: Bit-level confidence evolution during training across different hash code lengths (32, 64, and 128 bits) on Office-Home dataset. (Art → Real World task.)

**Sensitivity Analysis.** We conduct the sensitivity analysis to access the robustness of the proposed COLA with respect to the hyperparameter calibration set ratio $r_{cal}$. The analysis is performed on Office-Home with different hash code lengths. As Figure 6 shows, the retrieval performance of COLA remains stable across a wide range of calibration set ratios. We observe a better performance when $r_{cal}$ is set to 20%. More details are in Appendix D.3.

**Visualization.** To further understand the semantic structure of the learned representations, we adopt t-SNE visualization to demonstrate the discriminative binary codes on Office-Home. The results are shown in Figure 7. From the results, we can observe that our proposed COLA can generate more discriminative binary codes compared to the other two baseline methods.

## 5 CONCLUSION

To address the unreliable uncertainty handling in existing unsupervised domain adaptive hashing methods, this paper introduces COLA based on hierarchical conformal calibration. Our approach abandons traditional heuristic strategies in favor of a principled mechanism with rigorous statistical guarantees that quantifies uncertainty at both semantic and representation levels, thereby generating more reliable supervision signals for the target domain and modeling the stability of each hash bit. Extensive experiments on multiple benchmark datasets validate the superiority of our method, demonstrating consistent and significant improvements over state-of-the-art approaches. In summary, COLA provides a more reliable and adaptive solution for cross-domain retrieval tasks through systematic utilization of uncertainty, establishing a new paradigm that transforms uncertainty from an obstacle into a valuable resource for robust domain adaptive hashing.

## ETHICS STATEMENT

Our research adheres to the ICLR Code of Ethics.The code and related materials will be appropriately released to ensure transparency and reproducibility of our work. All datasets used in this study are publicly available.

## ACKNOWLEDGMENT

Ming Zhang, Jinsheng Huang and Junyu Luo are supported by grants from the National Key Research and Development Program of China with Grant No. 2023YFC3341203 and the National Natural Science Foundation of China (NSFC Grant Number 62276002).

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

## A    ACKNOWLEDGMENTS OF LLM USAGE

We utilized a large language model to aid our writing process, specifically for correcting grammar, improving sentence structure, and fetching related papers. The scientific contributions remain entirely our own.

## B    DATASET DETAILS

We evaluate our method on three widely-used cross-domain benchmarks for unsupervised domain adaptation tasks.

- *Office-Home* (Venkateswara et al., 2017): This dataset contains four distinct domains: Artistic (Ar), Clip Art (Cl), Product (Pr), and Real-World (Re). To ensure a fair comparison with previous work, we follow the standard protocol and establish six cross-domain image retrieval tasks among these domains, including: Pr→Re, Cl→Re, Re→Ar, Re→Pr, Re→Cl, Ar→Re.

- *Office-31* (Saenko et al., 2010): This dataset contains 31 categories from three domains: Amazon (Am), Webcam (We), and DSLR (Ds), with a total of over 4000 images. We similarly set up 6 image retrieval transfer tasks on this dataset: Am→Ds, Am→We, We→Ds, Ds→Am, We→Am, Ds→We.

- *Digits*: For handwritten digit recognition, we utilize the two classic datasets, MNIST (LeCun et al., 1998) and USPS (Hull, 1994). By alternating them as the source and target domains, we constructed 2 transfer tasks: MNIST→USPS and USPS→MNIST.

## C    BASELINE DETAILS

To comprehensively evaluate our COLA, we selected a series of state-of-the-art domain-adaptive hashing algorithms as comparative baselines, covering both unsupervised and adaptive hashing categories. To ensure fairness in comparison, the experimental results of all baseline methods were reproduced to match the reported results in their original publications. The core ideas of the baseline methods are briefly summarized as follows.

- *ITQ (Gong et al., 2012)*: A simple yet efficient alternating minimization algorithm with both supervised and unsupervised learning paradigms.

- *OCH (Liu et al., 2018)*: Approximates ordinal relations by a tensor ordinal graph, and employs ordinal constraint projection with a small set of centroids.

- *DSH (Jin et al., 2013)*: A variant of locality-sensitive hashing (LSH), which employs random projections to generate multi-view representations for metric learning.

- *SGH (Jiang & Li, 2015)*: Designed to transform high-dimensional features into binary codes, well-suited for large-scale semantic similarity mining tasks.

- *GraphBit (Wang et al., 2022)*: Explores bit-level interactions among features in continuous space, substantially alleviating the expensive search costs arising from training convergence difficulties in reinforcement learning.

- *GTH-g (Zhang et al., 2019)*: Finds the optimal hashing mapping functions for the target domain based on source-domain samples.

- *PWCF (Huang et al., 2020)*: Leveraging a Bayesian model for learning discriminative hash codes and infers the similarity structure through histogram features.

- *DHLing (Xia et al., 2021a)*: Optimizes hash codes through learnable clustering, and introduces a memory-bank mechanism to mitigate the effects of domain shift.

- *DAPH (Huang et al., 2021)*: Learning domain-invariant feature projections, which effectively reducing distribution discrepancies.

- *PEACE (Wang et al., 2023a)*: Applies pseudo-labeling techniques to learn target semantics, and subsequently minimizes domain transfer through implicit and explicit strategies.

- *DANCE (Wang et al., 2023b)*: A dual-level hashing learning framework that optimizes cross-domain high-level feature prototypes via contrastive learning.

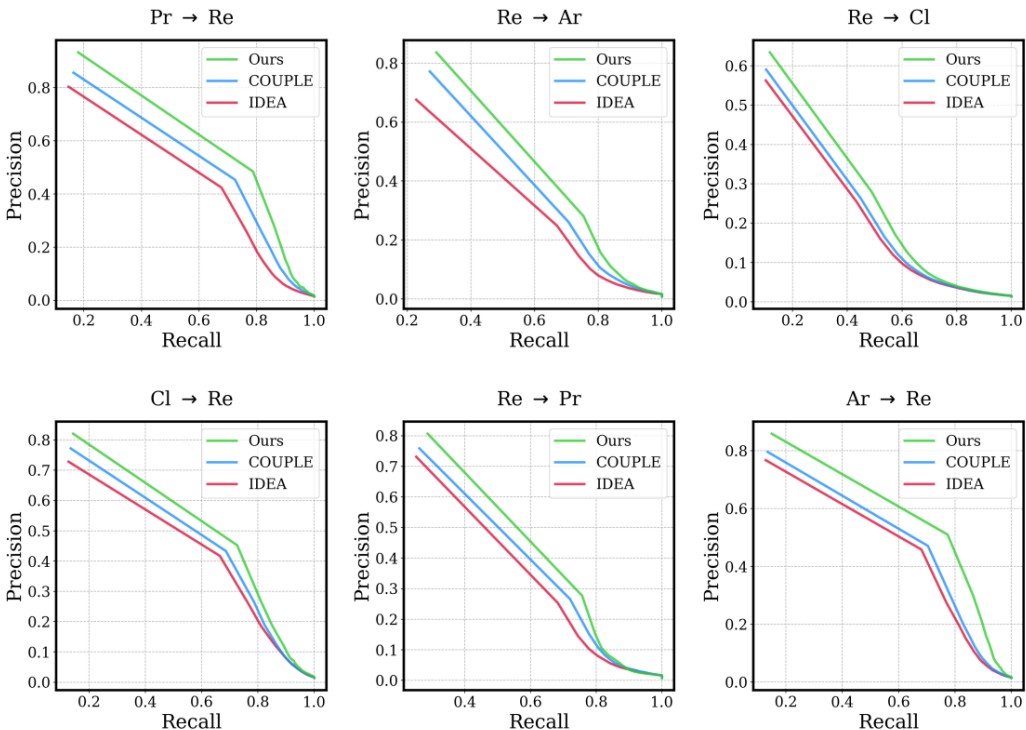

Figure 9: The comparison of precision-recall curves with 64-bit hash code on the Office-Home dataset.

- **IDEA** (*Wang et al., 2023d*): Decomposes visual representations into causal features which carry curical semantics and non-causal features, and produces binary codes from the causal components.
- **COUPLE** (*Luo et al., 2025*): Simulates the dynamic process via graph flow diffusion, and employs hierarchical mixup to achieve progressive cross-domain alignment.

## D    MORE EXPERIMENT RESULTS

### D.1    PERFORMANCE COMPARISON

To gain deeper insights of our COLA, we also conduct qualitative analysis experiments. We compared various approaches via precision-recall curves, Top-N precision curves, and Top-N recall curves. The results are shown in Figure 9, 10, and 11.

From the precision-recall curves in Figure 9, our COLA consistently outperforms all baseline methods across different cross-domain tasks on the Office-Home dataset. The curves demonstrate that COLA maintains higher precision values at all recall levels, indicating superior retrieval quality. Notably, the area under the PR curves for our method is significantly larger than that of competing approaches, suggesting more robust performance across varying similarity thresholds. The Top-N precision analysis in Figure 10 reveals that COLA achieves the highest precision scores across different values of N. This improvement is particularly pronounced when N is small, which is crucial for practical retrieval applications where users typically focus on top-ranked results. The consistent performance advantage across all six cross-domain tasks demonstrates the generalizability of our uncertainty calibration mechanism. Similarly, the Top-N recall curves in Figure 11 show that our method achieves superior recall rates compared to baseline approaches. The faster convergence of recall curves indicates that COLA can retrieve more relevant items within smaller candidate sets, which is essential for efficient large-scale retrieval systems. The substantial improvement margins across different domain adaptation scenarios validate the effectiveness of our conformal prediction-based calibration strategy in handling domain shift challenges.

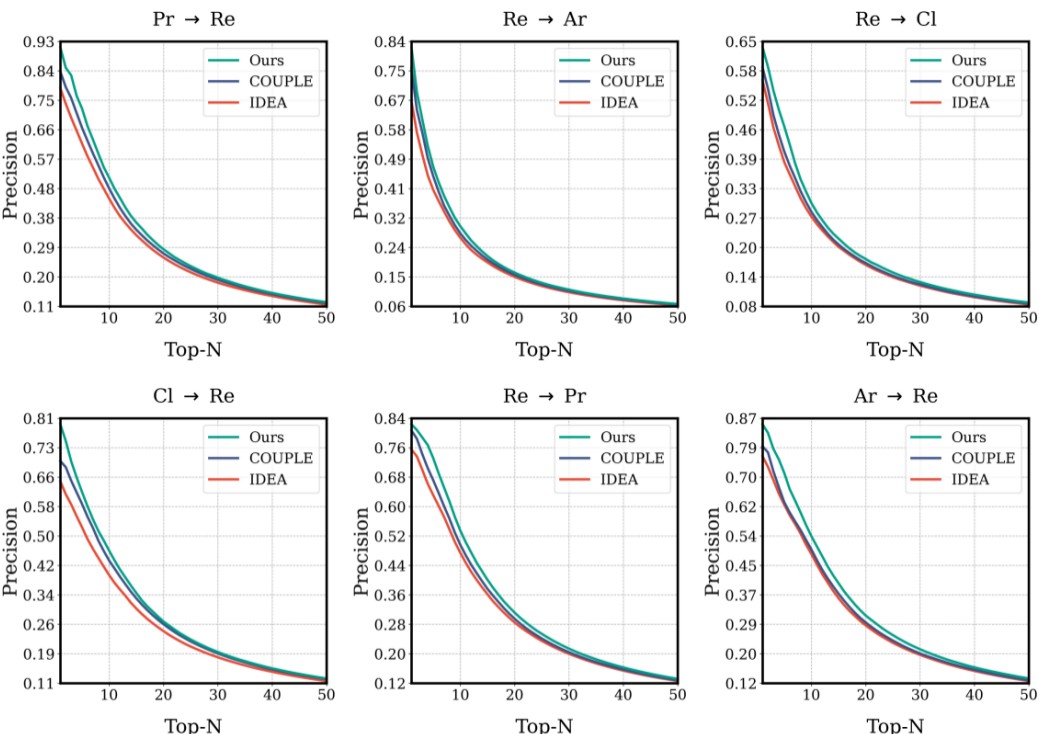

Figure 10: The comparison of Top-N precision curves with 64-bit hash code on Office-Home.

Table 4: The comparison of model variants on the Office-Home with 64-bit hash code.

| Variants | SC | RC | SR | Pr→Re | Cl→Re | Re→Ar | Re→Pr | Re→Cl | Ar→Re | Avg. |
|---|---|---|---|---|---|---|---|---|---|---|
| COLA (None) | | | | 59.65 | 46.47 | 49.66 | 59.56 | 31.92 | 47.56 | 49.14 |
| COLA-SC | ✓ | | | 59.92 | 47.08 | 49.89 | 59.81 | 32.07 | 47.82 | 49.43 |
| COLA-RC | | ✓ | | 60.83 | 48.64 | 51.33 | 61.22 | 35.84 | 52.70 | 51.76 |
| COLA-SR | | | ✓ | 61.41 | 47.77 | 52.24 | 61.69 | 35.11 | 51.06 | 51.55 |
| COLA w/o SC | | ✓ | ✓ | 65.13 | 48.89 | 54.12 | 65.97 | 40.01 | 53.04 | 54.53 |
| COLA w/o RC | ✓ | | ✓ | 65.95 | 49.39 | 56.16 | 66.87 | 39.68 | 53.18 | 55.21 |
| COLA w/o SR | ✓ | ✓ | | 65.25 | 49.30 | 53.29 | 65.62 | 39.95 | 55.03 | 54.74 |
| COLA (Full Model) | ✓ | ✓ | ✓ | **67.04** | **52.65** | **57.23** | **67.88** | **41.71** | **57.35** | **57.31** |

## D.2 ABLATION STUDY

To investigate the effectiveness of the core components in COLA, we conduct comprehensive ablation studies by systematically removing or modifying key components from the full model. We define several ablation variants to analyze different aspects of our approach:

- **COLA(Full Model)**: The complete COLA with all proposed components including semantic-level calibration, representation-level calibration, and self-regulating mechanism.

- **COLA w/o SC**: Removes the conformal prediction-based semantic uncertainty quantification module, using standard pseudo-labeling without uncertainty estimation.

- **COLA w/o RC**: Excludes the bit-wise reliability modeling component, treating all hash bits equally without considering their individual confidence levels.

- **COLA w/o SR**: Disables the dynamic objective balancing mechanism, using fixed weights for different loss components throughout training.

- **COLA-SC**: Retains only the semantic-level calibration component while removing other modules.

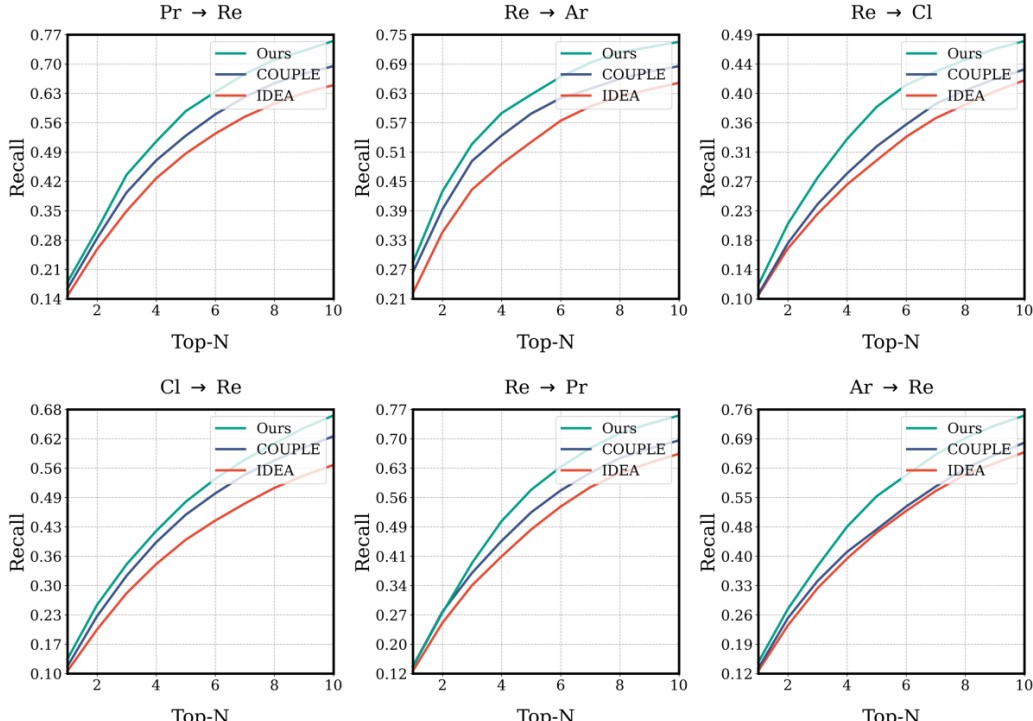

Figure 11: The comparison of Top-N recall curves with 64-bit hash code on Office-Home Dataset.

- **COLA-RC**: Keeps only the representation-level calibration while excluding semantic and self-regulating components.
- **COLA-SR**: Maintains only the dynamic balancing mechanism without uncertainty calibration modules.
- **COLA(None)**: Removes all three core components, which reduces to a standard deep unsupervised domain adaptive hashing baseline that relies solely on basic source supervision, a standard quantization loss, and an unweighted domain alignment loss.

Table 3 reports the comprehensive retrieval performance comparison of these ablation variants on the Office-Home dataset across all six cross-domain tasks. The experimental results provide several important insights:

**Impact of Semantic-Level Calibration**: Removing the conformal prediction-based semantic calibration leads to the most significant performance degradation, with average mAP dropping from 57.31% to 54.53%. This substantial decrease demonstrates the critical importance of uncertainty quantification in generating reliable soft pseudo-labels. Without proper semantic uncertainty estimation, the model struggles to distinguish between confident and uncertain predictions, leading to noisy supervision signals that harm domain alignment effectiveness.

**Importance of Representation-Level Calibration**: Excluding the bit-wise reliability modeling results in a notable performance decline to 55.21% average mAP. This confirms that not all hash bits contribute equally to the final representation quality, and modeling individual bit confidence is essential for robust cross-domain hashing. The representation-level calibration enables the model to focus on reliable bits while suppressing unreliable ones during the learning process.

**Effectiveness of Self-Regulating Mechanism**: Disabling the dynamic objective balancing leads to a performance drop to 54.74% average mAP. This validates the importance of adaptively adjusting the learning objectives based on real-time uncertainty estimates. The self-regulating mechanism prevents the model from over-fitting to uncertain predictions and ensures stable training dynamics across different domain adaptation scenarios.

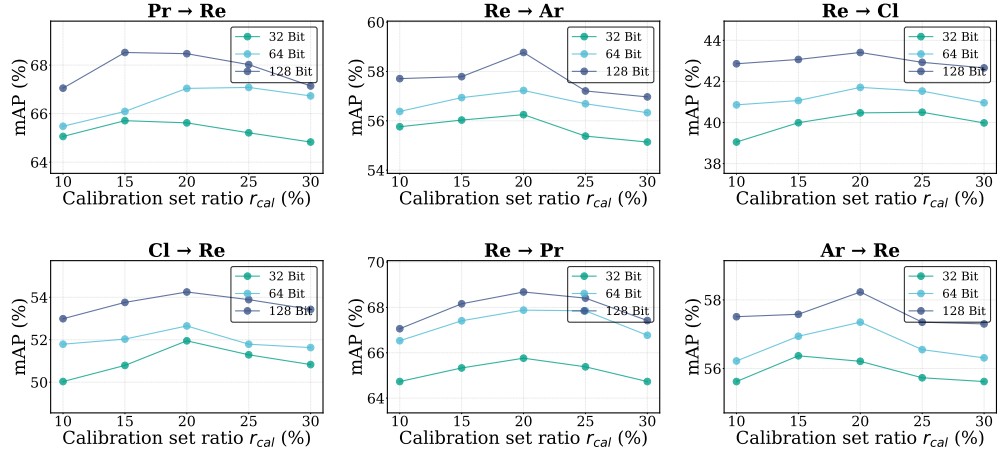

Figure 12: Sensitivity analysis on calibration set ratio $r_{cal}$ on Office-Home with different hash code lenghts.

Table 5: Retrieval time cost (ms) varies with code length.

|  | 16 Bit | 32 Bit | 48 Bit | 64 Bit | 96 Bit | 128 Bit |
|---|---|---|---|---|---|---|
| *Dense* Vector | 440.3 | 493.0 | 547.0 | 605.3 | 659.7 | 700.3 |
| Vanilla *Hash* Code | 15.38 | 17.94 | 20.32 | 18.97 | 21.83 | 22.37 |
| *UWHD* | 15.94 | 18.55 | 21.00 | 19.63 | 22.59 | 23.20 |
| Speed Up | 27.62× | 26.58× | 26.05× | 30.83× | 29.20× | 30.19× |

**Individual Component Analysis**: The variants with only single components (Only Semantic: 53.89%, Only Representation: 53.12%, Only Self-Regulating: 52.95%) all perform significantly worse than the full model, indicating that the synergistic combination of all components is crucial for optimal performance. Each component addresses different aspects of the cross-domain hashing challenge, and their integration creates a more robust and effective framework.

These ablation results conclusively demonstrate that each proposed component contributes meaningfully to the overall performance, and their combination in COLA achieves the best balance between uncertainty calibration and cross-domain adaptation effectiveness.

### D.3  SENSITIVITY ANALYSIS

We conduct the sensitivity analysis on the hyperparameter calibration set ratio $r_{cal}$. The analysis is performed on the Office-Home dataset with different hash code lengths. Figure 12 shows all experimental results of sensitivity analysis.

### D.4  CASE STUDY

We perform hash-based retrieval and present the top-5 results in Figure 13. COLA achieves higher retrieval accuracy than advanced baselines, validating the effectiveness of our proposed COLA and benefiting downstream retrieval-based tasks.

### D.5  SPEED TEST

In this part, we conducted a speed evaluation COLA and dense vector retrieval. Following previous works (Luo et al., 2025), we use a database of $10^6$ items. Each method was run $10^3$ times. We report the average retrieval time (ms) in Table 5. The results indicate that COLA could achieve substantially faster retrieval than dense vectors, and the inference-time calibration will not affect the efficiency, underscoring COLA's efficiency in large-scale retrievals. Since the inference-time

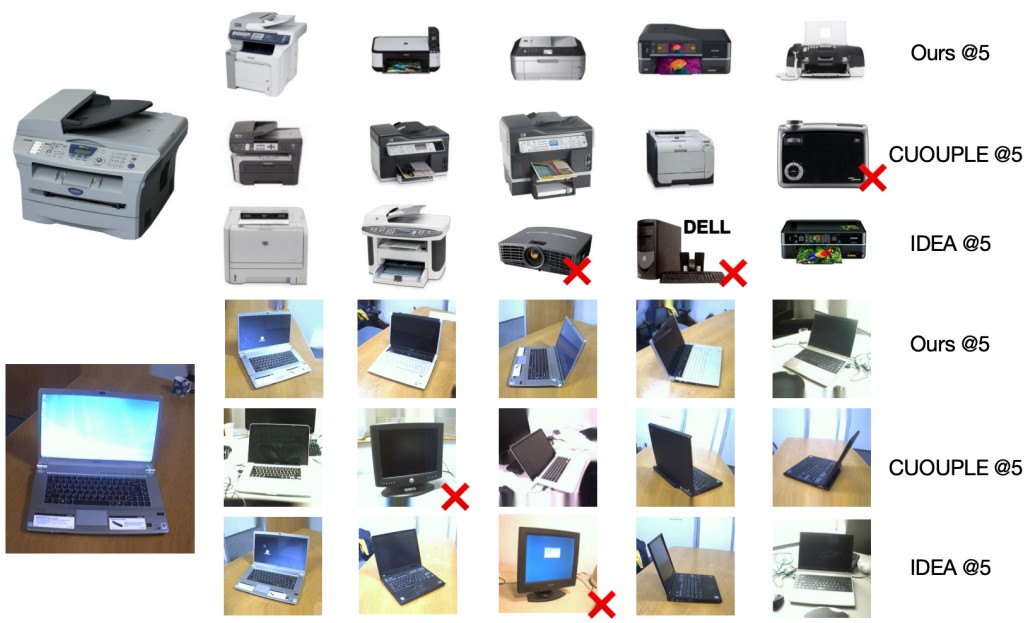

Figure 13: Case study on COLA, COUPLE and IDEA. Query the top 5 images on the Office-31 with 64 bits hash code.

metric is binarized into masked Hamming distance, it remains compatible with standard hardware-accelerated bitwise operations and existing ANN indexing structures.

## D.6 CALIBRATION STRATEGY ANALYSIS

To validate our calibration set construction, we compared our target centroid-based strategy with random, per-class, and density-aware sampling on Office-Home. As shown in Table 6, our method achieves the lowest MMD (0.0025) and highest mAP, indicating that our $D_{cal}$ best approximates the target distribution $D_t$.

Table 6: Comparison of calibration set selection strategies on Office-Home (Ar→Re).

| Strategy | mAP (%) | MMD ($D_{\text{cal}}, D_t$) |
|---|---|---|
| Ours | **56.34** | **0.0025** |
| Random Sampling | 55.21 | 0.0087 |
| Per-Class Sampling | 55.67 | 0.0057 |
| Density-Aware (K-Means) | 55.85 | 0.0032 |

## D.7 DYNAMIC ALPHA ABLATION

We compared our dynamic $\alpha$ mechanism with a fixed $\alpha$ baseline. Table 7 shows that dynamic $\alpha$ consistently outperforms fixed $\alpha$ across all datasets, achieving higher mAP and better empirical coverage (closer to $1 - \alpha$). The EMA parameter 0.7 was chosen to balance stability and adaptability.

We further investigated the impact of the EMA smoothing parameter $\alpha_{\text{sm}}$. As shown in Table 8, $\alpha_{\text{sm}} = 0.7$ yields the best performance, providing an optimal balance between stability and adaptivity.

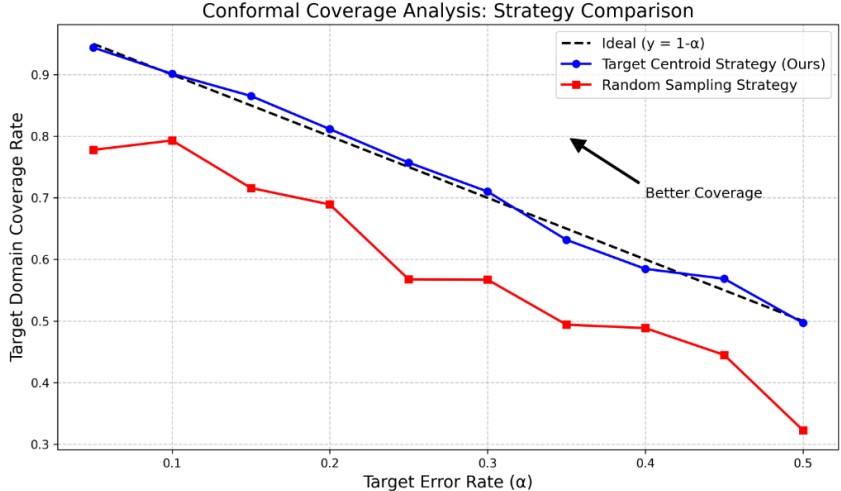

Figure 14: Conformal coverage analysis comparing our Target Centroid Strategy with Random Sampling. Our method (blue) closely follows the ideal coverage line ($y = 1-\alpha$), while random sampling (red) exhibits significant deviation.

Table 7: Ablation study of Fixed $\alpha$ vs. Dynamic $\alpha$.

| Dataset | mAP (%) | | Coverage | |
|---|---|---|---|---|
| | Fixed $\alpha$ | Dynamic $\alpha$ | Fixed $\alpha$ | Dynamic $\alpha$ |
| Office-Home | 55.22 | **57.31** | 0.87 | **0.91** |
| Office-31 | 66.43 | **67.11** | 0.88 | **0.93** |
| Digits | 69.57 | **70.41** | 0.91 | **0.94** |

### D.8 STANDARD HAMMING DISTANCE COMPARISON

To verify that our performance gain is not solely due to the weighted distance metric, we evaluated a variant *COLA (w/ Standard Hamming)* which uses the full model but retrieves with standard Hamming distance. As shown in Table 9, it still outperforms the best baseline COUPLE.

### D.9 THEORETICAL DISCUSSION

Theorem 3.1 indicates that the coverage guarantee depends on minimizing the TV distance between conformity score distributions. While directly computing this TV distance is intractable, our Representation Calibration (RC) serves as an effective empirical proxy. By enforcing bit stability, RC implicitly aligns the feature distributions of the source and target domains. As shown in our ablation study, the inclusion of RC significantly improves mAP (+2.62%), suggesting that it effectively reduces the distributional discrepancy and thus tightens the theoretical bound.

## E PROOF

Here we provide the detailed proof for Theorem 3.1. This theoretical guarantee of conformal prediction relies on the assumption that minimizing the feature distribution discrepancy (e.g., via MMD) effectively reduces the total variation distance between the conformity score distributions. This assumption holds approximately when the conditional distribution of conformal scores can be well approximated by a broad class of distribution families (Gretton et al., 2012). In such cases, the reduction of feature discrepancy implies the closeness of distributions. Conformal Prediction under covariate shift or distribution shift has been explored in prior work (Tibshirani et al., 2019; Barber et al., 2023; Guan, 2023). For a comprehensive introduction to conformal prediction, we refer readers to Angelopoulos & Bates (2021).

Table 8: Impact of EMA smoothing parameter $\alpha_{\mathrm{sm}}$ on mAP (%).

| Dataset | $\alpha_{\mathrm{sm}} = 0.5$ | $\alpha_{\mathrm{sm}} = 0.6$ | $\alpha_{\mathrm{sm}} = 0.7$ | $\alpha_{\mathrm{sm}} = 0.8$ | $\alpha_{\mathrm{sm}} = 0.9$ |
|---|---|---|---|---|---|
| Office-Home | 57.22 | 57.28 | **57.31** | 56.97 | 56.93 |
| Office-31 | 66.83 | 67.03 | **67.11** | 66.71 | 66.56 |
| Digits | 69.98 | 70.34 | **70.41** | 70.33 | 70.13 |

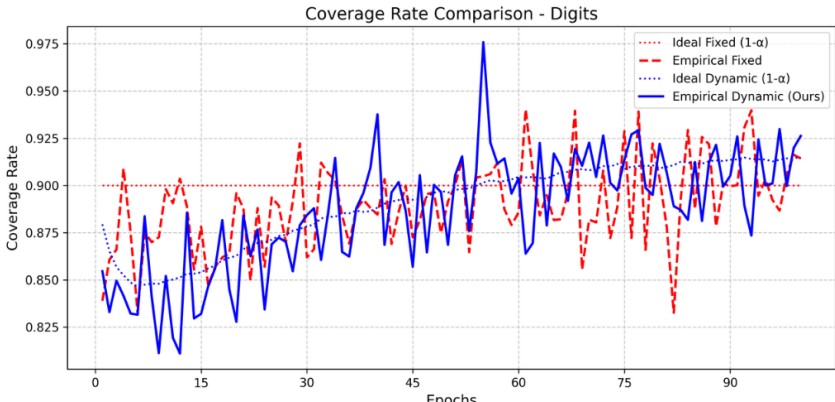

Figure 15: Coverage rate comparison on Digits dataset. Fixed $\alpha$ (red dashed) leads to under-coverage in early training, while our Dynamic $\alpha$ (blue solid) adaptively adjusts to maintain stable coverage around the ideal level.

**Theorem E.1** (Coverage Guarantee for Conformal Prediction under Domain Shift). *Let $d_{\mathrm{TV}}$ denote the total variation distance. Suppose $(X_{train}, Y_{train})$ and $(X_{test}, Y_{test})$ are random samples from the source and target distributions, respectively. Let $\hat{q}^{\mathrm{W}}$ be derived from equation 1. Then, the following coverage guarantee holds for the target domain:*

$$\mathbb{P}\left(s(X_{test}, Y_{test}) \leq \hat{q}^{\mathrm{W}}\right) \geq 1 - \alpha - d_{\mathrm{TV}}\left(s(X_{train}, Y_{train}), s(X_{test}, Y_{test})\right). \quad (12)$$

*If we further assume that the conformal score has a continuous distribution in both domains, then we also have the upper bound:*

$$\mathbb{P}\left(s(X_{test}, Y_{test}) \leq \hat{q}^{\mathrm{W}}\right) \leq 1 - \alpha + \frac{1}{n+1} + d_{\mathrm{TV}}\left(s(X_{train}, Y_{train}), s(X_{test}, Y_{test})\right). \quad (13)$$

*Proof.* By the coverage guarantee of conformal prediction on the source domain (Theorem 2.1 of Lei et al. (2018)), we have:

$$\mathbb{P}\left(s(X_{train}, Y_{train}) \leq \hat{q}^{\mathrm{W}}\right) \geq 1 - \alpha. \quad (14)$$

Recall that for any two random variables $U$ and $V$, the total variation distance is defined as

$$d_{\mathrm{TV}}(U, V) = \sup_{A \in \mathcal{F}} |\mathbb{P}(U \in A) - \mathbb{P}(V \in A)|. \quad (15)$$

Here $\mathcal{F}$ is the $\sigma$-algebra of measurable events. Now, consider the event $A = \{s \leq \hat{q}^{\mathrm{W}}\}$. Applying the definition of total variation, we immediately obtain:

$$\begin{aligned}
&\mathbb{P}\left(s(X_{test}, Y_{test}) \leq \hat{q}^{\mathrm{W}}\right) \\
=&\mathbb{P}\left(s(X_{train}, Y_{train}) \leq \hat{q}^{\mathrm{W}}\right) - \left(\mathbb{P}\left(s(X_{train}, Y_{train}) \leq \hat{q}^{\mathrm{W}}\right) - \mathbb{P}\left(s(X_{test}, Y_{test}) \leq \hat{q}^{\mathrm{W}}\right)\right) \\
\geq&1 - \alpha - d_{\mathrm{TV}}\left(s(X_{train}, Y_{train}), s(X_{test}, Y_{test})\right).
\end{aligned}$$
$$(16)$$

If we further assume that the conformal score has a continuous distribution in both domains, then by the upper bound for conformal prediction on the source domain (Lei et al., 2018), we have:

$$\mathbb{P}\left(s(X_{train}, Y_{train}) \leq \hat{q}^{\mathrm{W}}\right) \leq 1 - \alpha + \frac{1}{n+1}. \quad (17)$$

Table 9: Standard Hamming Distance Comparison on Office-Home (64 bits).

| Method | Avg mAP |
|---|---|
| COUPLE | 54.56 |
| COLA (w/ Standard Hamming) | 56.23 |
| COLA (Full w/ UWHD) | 57.31 |

Following the same logic as the proof of the lower bound, we can obtain the corresponding upper bound:

$$
\begin{aligned}
&\mathbb{P}\left(s(X_{test}, Y_{test}) \le \hat{q}^{\mathrm{W}}\right) \\
=&\mathbb{P}\left(s(X_{train}, Y_{train}) \le \hat{q}^{\mathrm{W}}\right) + \left(\mathbb{P}\left(s(X_{test}, Y_{test}) \le \hat{q}^{\mathrm{W}}\right) - \mathbb{P}\left(s(X_{train}, Y_{train}) \le \hat{q}^{\mathrm{W}}\right)\right) \\
\le& 1 - \alpha + \frac{1}{n+1} + d_{\mathrm{TV}}\left(s(X_{train}, Y_{train}), s(X_{test}, Y_{test})\right).
\end{aligned}
\tag{18}
$$

This completes the proof. □

