# OpenReview forum: "Conformalized Hierarchical Calibration for Uncertainty-Aware Adaptive Hashing"
_ICLR.cc/2026/Conference — ICLR 2026 Poster_

### Official Review · Reviewer_TTx6 · 2025-10-27

**Soundness:** 2
**Presentation:** 2
**Contribution:** 3
**Rating:** 6
**Confidence:** 3

**Summary:**

This paper introduces Conformal Hierarchical Calibration Adaptive Hashing, a framework for Unsupervised Domain Adaptive Hashing. The core problem it addresses is that existing UDAH methods suffer from noisy pseudo-labels and heuristic uncertainty handling, e.g., softmax thresholding, when adapting from a labeled source domain to an unlabeled target domain. At the semantic level, it uses conformal prediction set sizes to weight pseudo-label and alignment losses. At the representation level, it predicts individual hash bit stability to guide a weighted quantization loss and a novel weighted Hamming distance during retrieval. Experiments show COLA outperforms state-of-the-art methods on benchmark datasets.

**Strengths:**

The framework extends uncertainty quantification to the representation level by modeling the stability of individual hash bits. The proposed self-regulating mechanism intelligently uses quantified uncertainty signals to balance the multi-objective loss. The method achieves state-of-the-art performance, outperforming recent strong baselines on multiple benchmarks.

**Weaknesses:**

1. The paper's main solution adopts current theoretical results and methods in the literature of conformal prediction under domain shift, limiting its novelty.
2. The paper uses CP-based set size $1/|\mathcal{C}(x_t)|$ as its uncertainty measure. However, the paper does not compare it against a baseline that uses simpler heuristics (e.g., SoftMax threshold) within the COLA framework.
3. The retrieval efficiency could be low. Despite a comparison to dense vector in D.5, the current hash code weighted by floating points can show substantial time consumption compared to common binary hashing methods, violating the basic goal of hashing retrieval.
4. Formatting issues and typos:
 - Figure 5 obscures some text above it.
 - Typo around Line 231: "learnin"

**Questions:**

1. How does the domain shift severity influence the best threshold $r_{cal}$? This seems also heuristic.
2. Hashing retrieval are also designed for multi-label scenarios. Does the proposed method apply to them?

---

> ### Author Response · Authors · 2025-11-24
> **Thank you for your thorough review!**
>
> Dear Reviewer TTx6,
>
> Thank you for your thorough review and constructive feedback. We appreciate your valuable suggestions, which have helped us strengthen our paper. We will now address your comments point by point.

---

> ### Author Response · Authors · 2025-11-24
>
> > Q1. The paper's main solution adopts current theoretical results and methods in the literature of conformal prediction under domain shift, limiting its novelty.
>
> Thanks for your comment. While Conformal Prediction (CP) provides the theoretical foundation, we would like to highlight that COLA is not a mere application of CP to a new task. Our contributions lie in the domain-specific innovations for hashing retrieval:
>
> - **Bit-level Calibration**: General UDA methods focus on semantic labels. We propose a novel mechanism to quantify the stability of discrete hash bits. This addresses a unique challenge in hashing (optimization in Hamming space) that standard CP literature does not cover.
>
> - **Hierarchical Synergy**: We design a unified framework where the semantic-level CP guarantees filter out noise, creating a reliable supervision environment for learning the bit-level stability.
>
> - **Dynamic Self-Regulation**: We propose using the CP set size as a real-time feedback signal to dynamically balance the multi-objective optimization, which effectively solves the hyperparameter sensitivity issue in UDAH.
>
> We have revised the manuscript to explicitly emphasize these contributions.
>
> > Q2. The paper uses CP-based set size 1/|\mathcal{C}(x_t)| as its uncertainty measure. However, the paper does not compare it against a baseline that uses simpler heuristics (e.g., SoftMax threshold) within the COLA framework.
>
> Thank you for your comment. You are right. It is necessary to isolate the contribution of the CP-based measure.
>
> We have indeed conducted this comparison in our Ablation Study (Table 3). The variant _COLA w/o SC_ serves exactly as the baseline you suggested: it removes the Conformal Prediction module and relies on standard pseudo-labeling heuristics (using Softmax confidence) for semantic alignment, while keeping all other components (RC, SR) identical.
>
> **Table: Within-Framework Comparison on Office-Home**
> | Method | Uncertainty Measure | Avg mAP |
> | :--- | :--- | :---: |
> | **COLA w/o SC** | Standard Heuristic (Softmax) | 54.53% |
> | **COLA (Full)** | CP-based Set Size | **57.31%** |
>
> The results show that replacing the heuristic measure with our CP-based measure yields a significant improvement. This directly proves that the performance gain comes from the rigorous uncertainty quantification provided by Conformal Prediction, rather than just the model architecture itself.
>
> We have updated the discussion in manuscript.
>
> > Q3. The retrieval efficiency could be low. Despite a comparison to dense vector in D.5, the current hash code weighted by floating points can show substantial time consumption compared to common binary hashing methods, violating the basic goal of hashing retrieval.
>
> Thanks for raising this critical point. We would like to clarify that our method maintains the high efficiency of hashing retrieval:
>
> - **Binary Weights**: In the retrieval phase, we do not use floating-point weights. Instead, we binarize the learned bit-level confidence into $\{0, 1\}$ using a `round()` operation.
> - **Bitwise Operations**: Since the weights are binary, the Weighted Hamming Distance is calculated purely using efficient bitwise operations (XOR, AND, POPCNT), avoiding expensive floating-point arithmetic.
> - **Speed Verification**: As reported in the following table, our retrieval time is comparable to vanilla hashing and orders of magnitude faster than dense vector retrieval.
>
> | Method            | 16 Bit         | 32 Bit         | 48 Bit         | 64 Bit         | 96 Bit         | 128 Bit        |
> | ----------------- | -------------- | -------------- | -------------- | -------------- | -------------- | -------------- |
> | Dense Vector      | 440.3          | 493.0          | 547.0          | 605.3          | 659.7          | 700.3          |
> | Vanilla Hash Code | 15.38          | 17.94          | 20.32          | 18.97          | 21.83          | 22.37          |
> | UWHD (Ours)       | 15.94          | 18.55          | 21.00          | 19.63          | 22.59          | 23.20          |
> | Speed Up          | 27.62 $\times$ | 26.58 $\times$ | 26.05 $\times$ | 30.83 $\times$ | 29.20 $\times$ | 30.19 $\times$ |
>
> We have revised the Methodology and the Appendix.

---

> ### Author Response · Authors · 2025-11-24
>
> > Q4. Formatting issues and typos.
>
> Thanks for your careful reading. We have fixed the typo and adjusted the layout.
>
> > Q5. How does the domain shift severity influence the best threshold r_cal? This seems also heuristic.
>
> Thanks for the question. The parameter $r_{cal}$ selects source samples closest to the target domain to construct the calibration set.
>
> This selection is not purely heuristic but is grounded in the principle of **covariate shift adaptation**. By selecting source samples that are distributionally closer to the target domain (minimizing feature distance), we construct a calibration set that better satisfies the exchangeability assumption required by Conformal Prediction, thereby providing more reliable coverage guarantees on the target domain.
>
> While there is a trade-off (larger shift reduces the number of well-matched samples), our Sensitivity Analysis (Figure 5) demonstrates that the model's performance remains stable across a wide range of $r_{cal} \in [0.1, 0.3]$. This indicates that our method is robust to this hyperparameter and can effectively handle varying degrees of domain shift without sensitive tuning.
>
> > Q6. Hashing retrieval are also designed for multi-label scenarios. Does the proposed method apply to them?
>
> Thanks for the interesting question. Our current work focuses on the standard **single-label** Unsupervised Domain Adaptive Hashing (UDAH) setting.
> However, Conformal Prediction theory naturally supports multi-label outputs (as the prediction set can contain multiple labels). Therefore, our framework has the potential to be extended to multi-label hashing scenarios, which we consider a promising direction for future work.

---

> ### Author Response · Authors · 2025-11-24
>
> We are grateful for your constructive feedback. We have carefully revised the manuscript to address your concerns on novelty clarification, baseline comparison, retrieval efficiency, and multi-label extension potential. We hope these revisions are to your satisfaction!

---

### Official Review · Reviewer_eXJx · 2025-10-27

**Soundness:** 3
**Presentation:** 3
**Contribution:** 3
**Rating:** 4
**Confidence:** 3

**Summary:**

This paper proposes an uncertainty-aware deep hashing method for cross-domain retrieval, addressing the problem of domain shift through conformal prediction. The conformal prediction strategy transforms the predicted classes with the highest softmax probabilities into a pseudo-label set, which serves as a surrogate for potentially noisy ground-truth labels. The resulting confidence estimates of the hash codes are then used to guide domain alignment, hash learning, and retrieval. Experimental results demonstrate consistent improvements over existing baselines.

**Strengths:**

1. Presentation.
The paper is well-structured and clearly written, with a logical flow and well-defined sections. The mathematical notation is consistent and easy to follow, making the technical content accessible to readers familiar with the field.

2. Empirical Evaluation.
The experimental evaluation on benchmark datasets is comprehensive and convincing. The inclusion of detailed plots and visualizations effectively illustrates the results and provides clear insights into model behavior. The figures are well-designed and aesthetically appealing.

3. Originality and Significance.
While several prior studies have explored uncertainty-aware hashing strategies, the application of conformal prediction to deep hashing remains relatively underexplored. This work thus presents a novel and interesting direction that could inspire further research in uncertainty modeling for retrieval under domain shift.

**Weaknesses:**

1. Need for a Brief Introduction to Conformal Prediction.
The paper would benefit from a short subsection introducing the basic concepts and principles of conformal prediction. This would make the paper more self-contained and accessible to readers who may not be familiar with this framework.

2. Missing Discussion of Related Work
The literature review omits several recent works that have explored uncertainty-aware deep hashing or uncertainty modeling in retrieval, which are directly relevant to this study. In particular, the following papers should be cited and discussed for a more comprehensive positioning of the proposed approach:

- Warburg, Frederik, et al. "Bayesian triplet loss: Uncertainty quantification in image retrieval." Proceedings of the IEEE/CVF International conference on Computer Vision. 2021.

- Warburg, Frederik, et al. "Bayesian metric learning for uncertainty quantification in image retrieval." Advances in Neural Information Processing Systems 36 (2023): 69178-69190.

- Wang, Yucheng, and Mingyuan Zhou. "Uncertainty-aware unsupervised video hashing." The 26th International Conference on Artificial Intelligence and Statistics. PMLR, 2023.

- Wang, Yucheng, Mingyuan Zhou, and Xiaoning Qian. "Hashing with Uncertainty Quantification via Sampling-based Hypothesis Testing." Transactions on Machine Learning Research.

- Tang, Haomiao, et al. "Modeling uncertainty in composed image retrieval via probabilistic embeddings." Proceedings of the 63rd Annual Meeting of the Association for Computational Linguistics (Volume 1: Long Papers). 2025.

3. Some design choices lack sufficient motivation. Please refer to Question 3 for details.

4. Missing Complexity Analysis: Section 3.3.1: Uncertainty-aware Weighted Hamming Distance.
While the introduction of a weighted Hamming distance adds modeling flexibility, the paper does not provide a computational complexity analysis. This omission is problematic because the weighting mechanism changes the operational nature of the distance metric — from bitwise operations to floating-point vector arithmetic — which may significantly impact runtime performance and scalability in large-scale data retrieval. A formal complexity discussion, or at least empirical runtime analysis, is necessary to demonstrate the method’s feasibility in real-world retrieval settings.

5. Could the authors provide an ablation experiment in which all three components (SC, RC, and SR) are removed? This would help clarify the individual contribution and overall effect of each module.

**Questions:**

1. Line 144-146: "The size of these sets rigorously quantifies model uncertainty and is directly converted to weights that adaptively suppress the harmful effects of high-uncertainty samples in pseudo-label learning and domain alignment. " The authors should clarify why this uncertainty is categorized as model uncertainty rather than data uncertainty. Since the uncertainty appears to arise from the variability or ambiguity in the data samples (e.g., ambiguous labels or overlapping class features), it might be more appropriately described as data uncertainty. A brief justification or theoretical reasoning distinguishing the two types of uncertainty in this context would strengthen the conceptual rigor of the paper.

2. The paper employs a conformal prediction strategy to transform high-confidence softmax predictions into pseudo-label sets that replace the original noisy labels. However, at the early stages of training — when the model is randomly initialized — the softmax probabilities are effectively random. This implies that the generated pseudo-label sets could be even noisier than the original labels, potentially leading to unstable or misleading supervision.
Could the authors provide a discussion or empirical justification addressing this issue? For example, is there any mechanism (e.g., warm-up phase, confidence thresholding, or delayed pseudo-labeling) to mitigate the effect of unreliable predictions during early training?

3. Section 3.3: Proxy Task for Bit Stability. It appears that the stability measure $v_{i,k}$ could be obtained directly by adding perturbations (e.g., Gaussian noise) to the feature representation and simulating the corresponding bit responses. If that is the case, could the authors clarify why a separate confidence prediction head, $G_{bit}$, is necessary?

4. Experimental Setup and Fairness of Comparison
It is unclear whether the proposed model and the baseline models are evaluated using the proposed Uncertainty-Aware Weighted Hamming Distance (UWHD) or the standard Hamming distance. If the proposed method uses UWHD while the baselines rely on the original Hamming distance, this may introduce a fairness issue in performance comparison.
The authors may need to clarify this point and provide additional experiments where: (1) Baseline models are also evaluated using the proposed UWHD to ensure a fair comparison under consistent retrieval metrics. (2) An ablation study isolates the contribution of UWHD itself to quantify its effect on retrieval accuracy and runtime.
This clarification is important because the proposed UWHD may both increase retrieval cost and contribute independently to performance improvements.

---

> ### Author Response · Authors · 2025-11-24
> **Thank you for your thorough review and constructive feedback!**
>
> Dear Reviewer eXJx,
>
> Thank you for your thorough review and constructive feedback. We appreciate your valuable suggestions, which have helped us strengthen our paper. We will now address your comments point by point.

---

> ### Author Response · Authors · 2025-11-24
>
> > Q1. Add a brief introduction to Conformal Prediction in Preliminaries.
>
> Thanks for your constructive suggestion. We have added a Conformal Prediction Basics paragraph in Section 2 Preliminaries.
>
> ```latex
> Conformal Prediction Basics.
>
> Conformal prediction is a distribution-free framework that constructs prediction sets with rigorous statistical guarantees. Given a calibration set and a user-defined error rate $\alpha$, it produces a set $\mathcal{C}(x)$ for a new input $x$ such that the true label $y$ is contained in $\mathcal{C}(x)$ with probability at least $1-\alpha$. This coverage guarantee relies on the exchangeability of data, which we address in the domain adaptation setting via weighted conformal prediction.
> ```
>
> > Q2. Discuss missing related works on uncertainty-aware retrieval and hashing.
>
> Thank you for providing these references. We have carefully studied and incorporated them into our discussion. We have added a new subsection _Uncertainty in Retrieval_ to discuss them in detail:
>
> - **Probabilistic Embeddings:** Warburg et al. [1, 2] and Tang et al. [5] capture uncertainty by modeling embeddings as stochastic distributions.
> - **Generative Hashing:** Wang et al. [3, 4] utilize variational autoencoders or hypothesis testing to estimate hash code uncertainty.
>
> Our COLA method differs significantly: COLA is based on the distribution-free framework and provides coverage guarantees specifically for domain shift in domain adaptation scenarios, while avoiding the expensive sampling overhead of Bayesian methods, making it more suitable for efficient retrieval.
>
> We have updated the Related Works accordingly.
>
> ```latex
> Uncertainty in Retrieval.
>
> Recent works have explored uncertainty modeling in retrieval. [1,2] proposed Bayesian metric learning to model aleatoric and epistemic uncertainty via stochastic embeddings. [5] utilized probabilistic embeddings for composed image retrieval. In hashing, [3,4] introduced generative approaches to estimate hash code uncertainty. Unlike these model-based methods, our COLA employs a distribution-free conformal prediction framework. It provides coverage guarantees under domain shift and hierarchically calibrates uncertainty at both semantic and representation levels without expensive sampling.
> ```
>
> [1] Bayesian triplet loss: Uncertainty quantification in image retrieval. CVPR 2021.
>
> [2] Bayesian metric learning for uncertainty quantification in image retrieval. NeurIPS 2023.
>
> [3] Uncertainty-aware unsupervised video hashing. AISTATS 2023.
>
> [4] Hashing with Uncertainty Quantification via Sampling-based Hypothesis Testing. TMLR.
>
> [5] Modeling uncertainty in composed image retrieval via probabilistic embeddings. ACL 2025.
>
> > Q3. Justify the necessity of the separate bit stability prediction head.
>
> Thanks for your question. We employ a separate Bit Head instead of inference-time perturbation for two key reasons:
>
> - **Efficiency:** Direct perturbation requires multiple forward passes per query, largely increasing retrieval latency. Our lightweight Bit Head predicts stability in a single forward pass.
> - **Training Guidance:** The Bit Head provides a differentiable stability score $w^{bit}$ during training, allowing us to weight the quantization loss (Eq. 9).
>
> We have updated the Methodology accordingly.
>
> ```latex
> We employ a separate prediction head rather than on-the-fly perturbation during inference to ensure retrieval efficiency. Direct perturbation would require multiple forward passes per query, significantly increasing latency. Our lightweight head predicts stability in a single pass ($O(1)$), maintaining the speed advantage of hashing.
> ```

---

> ### Author Response · Authors · 2025-11-24
>
> > Q4. Provide complexity analysis and runtime comparison for UWHD.
>
> Thanks for your comment. We fully considered efficiency when designing UWHD and adopted the following measures:
>
> - **Binarized Implementation:** During retrieval, we binarize the predicted bit confidence to $\{0, 1\}$. This means UWHD is implemented as a Masked Hamming Distance, relying entirely on efficient bitwise operations and avoiding floating-point calculations.
> - **Speed Test:** We provided a detailed speed test in Appendix. Results show that UWHD retrieval speed (\~23.2ms) is comparable to vanilla hashing (\~22.4ms) and over **30x faster** than dense vector retrieval.
>
> | Method          |  16 Bit   |  32 Bit   |  48 Bit   |  64 Bit   |  96 Bit   |  128 Bit  |
> | :-------------- | :-------: | :-------: | :-------: | :-------: | :-------: | :-------: |
> | Dense Vector    |   440.3   |   493.0   |   547.0   |   605.3   |   659.7   |   700.3   |
> | Vanilla Hash    |   15.38   |   17.94   |   20.32   |   18.97   |   21.83   |   22.37   |
> | **UWHD (Ours)** | **15.94** | **18.55** | **21.00** | **19.63** | **22.59** | **23.20** |
>
> We have clarified the efficiency of UWHD in Methodology.
>
> ```latex
> In practice, we binarize the query weights $w_{q,k} \in \{0, 1\}$ via rounding, enabling UWHD to be computed via efficient bitwise operations. As shown in Appendix, our method achieves comparable speed to vanilla hashing.
> ```
>
> > Q5. Highlight the ablation study removing all components (SC, RC, SR).
>
> Thank you for your comment. We have explicitly highlighted this in the Ablation Study. The variant _COLA (None)_ in Table 3 represents the baseline with all three components (SC, RC, SR) removed, which serves as a standard UDAH baseline.
>
> **Table: Ablation studies on Office-Home (64 bits)**
> | Variants | Avg. mAP |
> | :--- | :---: |
> | **COLA (None)** | **49.14** |
> | COLA-SC | 49.43 |
> | COLA-RC | 51.76 |
> | COLA-SR | 51.55 |
> | COLA w/o SC | 54.53 |
> | COLA w/o RC | 55.21 |
> | COLA w/o SR | 54.74 |
> | **COLA (Full)** | **57.31** |
>
> Results show that our full model (Avg 57.31%) significantly outperforms _COLA (None)_ by +8.17\%, demonstrating the substantial cumulative contribution of our proposed modules.
>
> We have updated the Ablation Study accordingly.
>
> ```latex
> The variant \textit{COLA (None)} in Table~\ref{tab:ablation_office_home} represents the baseline with all three components (SC, RC, SR) removed, which serves as a standard UDAH baseline.
> ```
>
> > Q6. Clarify the type of uncertainty (Model vs. Data vs. Predictive).
>
> Thanks for your question. The size of the conformal prediction set quantifies Predictive Uncertainty, which encompasses both Model Uncertainty (Epistemic, due to domain shift) and Data Uncertainty (Aleatoric, due to sample ambiguity). In UDAH, we use this aggregate uncertainty to identify samples the model cannot reliably predict. We have updated the terminology in Methodology to *predictive uncertainty* for precision.
>
> We have updated the Methodology accordingly.
>
> ```latex
> The size of this prediction set naturally and rigorously quantifies the predictive uncertainty (encompassing both aleatoric and epistemic uncertainty) for each sample.
> ```

---

> ### Author Response · Authors · 2025-11-24
>
> > Q7. Explain how early training instability is handled (Self-Regulating Mechanism).
>
> Thanks for your question. Our Self-Regulating Mechanism naturally acts as an **automatic warm-up strategy**.
>
> - **Automatic Suppression:** In the early training stages, the model is uncertain, resulting in large prediction sets and low semantic confidence $\bar{w}^{\mathrm{sem}}$. Consequently, the target loss weight $\lambda_{\mathrm{target}}$ (Eq. 11) is automatically suppressed. This prevents the model from overfitting to noisy pseudo-labels early on.
> - **Progressive Adaptation:** As the model learns from the source domain, uncertainty decreases, and $\lambda_{\mathrm{target}}$ gradually increases, smoothly engaging the domain adaptation process.
>
> We have updated the Methodology to emphasize the warm-up strategy.
>
> ```latex
> This mechanism naturally acts as a warm-up strategy: early in training, high uncertainty leads to low $\lambda_{\mathrm{target}}$, preventing the model from overfitting to noisy pseudo-labels. As the model learns from the source domain, uncertainty decreases, and the target adaptation gradually engages.
> ```
>
> > Q8. Address fairness of comparison and isolate UWHD contribution.
>
> Thanks for your question. To address your concerns, we added a new ablation variant _COLA (w/ Standard Hamming)_, which uses the trained COLA model but retrieves with standard Hamming distance.
>
> **Table: Fairness Comparison on Office-Home (64 bits)**
> | Method | Avg mAP |
> | :--- | :---: |
> | Best Baseline (COUPLE) | 54.56 |
> | **COLA (w/ Standard Hamming)** | **56.23** |
> | **COLA (Full w/ UWHD)** | **57.31** |
>
> Results on Office-Home (64 bits) show:
>
> - **COLA (w/ Standard Hamming)** achieves 56.23% mAP, still outperforming the best baseline COUPLE (54.56%). This proves our training framework learns superior hash codes even without the weighted distance.
> - **COLA (Full w/ UWHD)** further improves to 57.31%, quantifying the specific contribution of UWHD.
>
> We have updated the Appendix accordingly.

---

> ### Author Response · Authors · 2025-11-24
>
> We are grateful for your constructive feedback. We have carefully revised the manuscript to address your concerns on conformal prediction basics, related work, efficiency clarification, ablation study, and fairness comparison. We hope these revisions are to your satisfaction!

---

### Official Review · Reviewer_uPy1 · 2025-10-29

**Soundness:** 2
**Presentation:** 3
**Contribution:** 2
**Rating:** 6
**Confidence:** 4

**Summary:**

This paper proposes a conformalized hierarchical calibration method called COLA for unsupervised domain adaptive hashing that integrates uncertainty quantification at both semantic and representation levels. The method leverages conformal prediction to produce confidence-calibrated pseudo-labels and introduces bit-level reliability modeling for hash code robustness. Additionally, a self-regulating mechanism dynamically balances learning objectives based on real-time uncertainty. Experiments on some benchmarks demonstrate improvement over state-of-the-art baselines.

**Strengths:**

1.	The use of conformal prediction for uncertainty estimation in domain adaptive hashing is innovative and grounded in theory, supported by formal coverage guarantees under domain shift.

2.	The hierarchical semantic and bit-level calibration is well-structured and addresses both label noise and representation robustness.

3.	Extensive experiments and clear performance gains across diverse benchmarks demonstrate effectiveness and robustness.

**Weaknesses:**

1.	The method includes introduces multiple components inducing semantic-level conformal calibration, dynamic threshold, bit-level reliability modeling, and a self-regulating optimization mechanism. While each is well-motivated, it would be beneficial if the authors could more explicitly clarify which part constitutes the core contribution and novelty of the method.

2.	The model contains many weighting mechanisms such as semantic-level sample weights, batch-level confidence weights, bit-level stability weights, and dynamically regulated loss coefficients. While each seems intuitively reasonable, the true necessity and effectiveness of these weights remain unclear. Although the conformal calibration module is theoretically grounded, the other weighting mechanisms also appear heuristic.

3.	Some notations and equations need further explanations. For example, how to compute $\hat{q}^W$ in Eq. 1 is unclear.

**Questions:**

see weaknesses.

---

> ### Author Response · Authors · 2025-11-24
> **Thank you for your thorough review!**
>
> Thank you for your thorough review and constructive feedback. We appreciate your valuable suggestions, which have helped us strengthen our paper. We will now address your comments point by point.

---

> ### Author Response · Authors · 2025-11-24
>
> > W1. Core Contribution and Novelty
>
> Thanks for your valuable comment. The core innovation of COLA is the hierarchical conformal calibration framework that quantifies
> uncertainty at two complementary levels with rigorous statistical guarantees. This framework consists of:
>
> - **Semantic-Level Calibration:** We replace risky point predictions with conformal prediction sets, whose sizes naturally quantify semantic uncertainty with coverage guarantees. This is theoretically grounded and provides statistically valid uncertainty estimates even under domain shift. The prediction set size directly informs sample-wise and batch-wise weighting strategies for pseudo-labeling and domain alignment.
>
> - **Representation-Level Calibration:** We model bit-wise stability through a sign-consistency proxy task, yielding fine-grained reliability scores for each hash bit. This extends uncertainty quantification from semantic labels to the representation space, enabling uncertainty-aware weighted Hamming distance during retrieval—a novel contribution beyond existing hashing methods.
>
> - **Auxiliary Mechanisms:** The dynamic threshold adjustment and self-regulating mechanism serve as supporting components that dynamically adapt the learning process based on the quantified uncertainty from the two core calibration levels.
>
> To our knowledge, this is the first work to systematically introduce conformal prediction theory into domain adaptive hashing and extend uncertainty quantification to the bit level. Existing methods rely on heuristic confidence scores without statistical guarantees, while our framework provides principled uncertainty estimates that drive both training and inference.
>
> We have enhanced the discussion in Introduction and Methodology accordingly.
>
> > W2. Necessity and Effectiveness of Weighting Mechanisms
>
> Thanks for your comment. We categorize our weights into three types with different levels of theoretical support:
>
> - **Theoretically-Grounded Weights: Semantic-Level Calibration.** The semantic-level sample weights $w_t^{\mathrm{sem}} = 1/|\mathcal{C}(x_t)|$ are directly derived from conformal prediction theory. The prediction set size $|\mathcal{C}(x_t)|$ provably quantifies model uncertainty with statistical coverage guarantees (Theorem 1). This weight is not heuristic but follows naturally from rigorous statistical theory. Smaller prediction sets indicate higher certainty, and their reciprocal serves as a principled confidence measure.
>
> - **Theoretically-Motivated Weights: Bit-Level Calibration.** The bit-level stability weights are motivated by a sign-consistency principle: a reliable bit must maintain its sign under minor perturbations. If a bit flips easily under small input perturbations, it cannot be trusted for stable retrieval. This proxy task is not arbitrary but follows naturally from the quantization objective.
>
> - **Empirically-Validated Weights:** The necessity of these mechanisms is empirically validated by our ablation study (Table 3). As shown below, removing the semantic weights (w/o SC) leads to a -2.78\% drop in mAP, and removing the bit-level weights (w/o RC) leads to a -2.10\% drop. These significant performance declines quantitatively prove that each weighting component is not just heuristic but essential for the final performance.
>
> **Table: Impact of Weighting Mechanisms on Office-Home**
> | Variant | Avg mAP |
> | :--- | :---: |
> | COLA (Full) | 57.31% |
> | w/o Semantic Weights (SC) | 54.53% |
> | w/o Bit Weights (RC) | 55.21% |
>
> We have enhanced the discussion in Methodology accordingly.
>
> > W3. Notation and Computation of $\hat{q}^{\mathrm{W}}$
>
> Thanks for your valuable comment. The weighted quantile threshold $\hat{q}^{\mathrm{W}}$ is computed as follows: we collect all conformity scores $\{ s( x_i, y_i ) \}\_{i=1}^{n_{\mathrm{cal} } } $ from the calibration set, sort them in ascending order, and select the $\lceil(n_{\mathrm{cal}}+1)(1-\alpha_t)\rceil$-th smallest score as the threshold. The superscript `W` denotes that this quantile is weighted by the dynamic miscoverage level $\alpha_t$, which adapts based on model performance rather than remaining fixed.
>
> We have added this details in Methodology accordingly.

---

> ### Author Response · Authors · 2025-11-24
>
> We are grateful for your constructive feedback. We have carefully revised the manuscript to address your concerns on core contribution, weighting mechanisms, and notation clarity. We hope these revisions are to your satisfaction!

---

### Official Review · Reviewer_bmwE · 2025-10-31

**Soundness:** 3
**Presentation:** 3
**Contribution:** 3
**Rating:** 4
**Confidence:** 4

**Summary:**

This paper addresses the problem of uncertainty estimation in unsupervised domain adaptive hashing (UDAH). It introduces COLA, a hierarchical framework that unifies semantic-level conformal calibration and representation-level (bit-wise) reliability modeling under a self-regulating training scheme. At the semantic level, COLA employs conformal prediction to generate multi-label prediction sets instead of single pseudo-labels, ensuring controlled coverage even under domain shift. At the representation level, COLA estimates per-bit reliability by measuring feature stability under perturbations. These bit confidences reweight quantization loss and are also applied during retrieval as a query-weighted Hamming distance, enabling uncertainty-aware retrieval. Extensive experiments on Office-Home, Office-31, and MNIST-USPS datasets show consistent improvements in mAP and robustness, supported by ablations for each module.

**Strengths:**

- A clear strength of this paper is its two-level uncertainty design, which handles noise both in pseudo-label assignment and binary code learning. The semantic-level conformal predictor generates calibrated prediction sets to control pseudo-label confidence, while the representation-level reliability estimator measures bit-level stability and integrates these confidences into the learning process. This combination enables consistent management of uncertainty from prediction to representation.

- The paper provides an explicit theoretical framing via Theorem 3.1, which establishes a coverage bound of 1 - \alpha_t - \text{TV}(P_s, P_t) under domain shift. The theorem connects conformal calibration with target-domain generalisation, grounding the semantic calibration step in probabilistic coverage terms. Although the reduction of the TV term has not been empirically proven, the presence of an analytical coverage expression distinguishes the paper from heuristic uncertainty-weighting works.

- The inclusion of a self-regulating mechanism that reweights alignment and quantisation losses according to aggregated confidence metrics represents another notable design feature. By dynamically adjusting these weights, the framework avoids the need for manual tuning of trade-off hyperparameters and adapts its learning emphasis based on reliability feedback.

**Weaknesses:**

1) **Retrieval efficiency and scalability untested.** The proposed uncertainty-weighted Hamming distance (Eq. 9) replaces the standard bitwise XOR + popcount kernel that underpins hashing’s speed with a per-bit weighted sum. While accuracy gains are clearly demonstrated on Office-Home, Office-31, and Digits benchmarks, these datasets are only moderate in size and do not represent million-scale retrieval scenarios. The paper does not report retrieval latency, throughput, or index compatibility for this weighted metric, and its complexity analysis covers only calibration (sorting O(n_{\text{cal}}\log n_{\text{cal}})). Thus, the impact on large-scale system efficiency remains unquantified.

This is because the proposed distance fundamentally changes the computation model from binary bit-operations to floating-point per-bit weighting. Therefore, it alters the very assumption that gives hashing its appeal: sub-millisecond similarity search using bitwise logic. Demonstrating that the new retrieval kernel still scales efficiently is therefore not an auxiliary test but a technical validation step. It confirms that COLA preserves hashing’s defining property of efficiency while adding uncertainty awareness. Without such timing and scalability evidence, the proposed method’s theoretical soundness is intact, yet its practical completeness is uncertain. At least, it is unclear to me whether the algorithm remains a true hashing framework or behaves more like a compact continuous embedding model.

Adding runtime and scalability experiments on standard large-scale retrieval corpora such as NUS-WIDE, MS-COCO etc, along with a UDA-at-scale check on VisDA-2017 or DomainNet would directly verify that the uncertainty-weighted Hamming distance maintains the computational advantages of hashing.


2. **Calibration-set construction is heuristic.**
The paper defines D_{\text{cal}} as the top r_{\text{cal}} of source samples nearest to the target centroid (default 20%). While this is a reasonable heuristic and sensitivity to r_{\text{cal}} is analyzed, there is no comparison with alternative calibration-set selection rules (e.g., per-class, random, or density-aware sampling). Since Theorem 3.1’s coverage guarantee assumes that the calibration distribution approximates the target-domain score distribution, could the authors clarify how representative the chosen D_{\text{cal}} is of the target domain in practice? Have the authors tested whether alternative calibration-set constructions produce similar coverage and retrieval performance, or could provide evidence (e.g., distributional similarity or coverage plots) to verify that the conformal bound remains valid under this heuristic?

3. The adaptive \alpha_t is an elegant idea, but its contribution is not explicitly ablated against a fixed-\alpha baseline. That is, the paper introduces an EMA-based dynamic \alpha_t, but there is no analysis of why a time-varying \alpha_t yields better or more stable coverage than a fixed \alpha, nor is there an ablation comparing the two. Since \alpha_t directly controls prediction-set size and, through it, the confidence-weighted alignment, please provide: (i) a fixed-\alpha baseline on all three datasets; (ii) a plot of empirical coverage vs. time for both fixed and dynamic \alpha; and (iii) an explanation of how the EMA parameters were chosen. Without this, the dynamic schedule appears ad hoc rather than necessary.

4. The theoretical result (Theorem 3.1) depends on a reduction of the TV distance between source and target conformity-score distributions through alignment. However, the paper does not empirically estimate or proxy this TV term, so the connection between the alignment loss and actual coverage improvement remains unverified experimentally. Please (i) justify why this bound is meaningful under the non-exchangeable source/target feature distributions in these datasets, and (ii) provide an empirical proxy (e.g., pre-/post-alignment MMD on the conformity scores) to support the claim that alignment makes the conformal coverage guarantee non-vacuous.

**Questions:**

1. The current ablation toggles semantic calibration (SC), representation calibration (RC), and self-regulation (SR), and concludes that each helps. However, this does not isolate which part of the proposed conformalization is responsible for the final mAP improvements, because all components are evaluated under the same centroid-based D_{\text{cal}} and dynamic \alpha_t. To make the causal story convincing, please provide an ablation that (i) fixes D_{\text{cal}} to a random subset and (ii) fixes \alpha, and then measures the marginal gain from adding SC and RC separately. This will show whether the benefit truly comes from conformal calibration, rather than from generic confidence weighting.

2. All reported experiments are on Office-Home, Office-31, and MNIST↔USPS, which are small-to-moderate in size. Yet the method introduces a retrieval rule that is more expensive than standard binary hashing. Please clarify the intended scope of the claim: is COLA meant only for moderate-scale UDAH (as in the current experiments), or is it intended to be a scalable, uncertainty-aware hashing method? If the latter, additional experiments on larger-scale benchmarks (DomainNet, VisDA-2017, or NUS-WIDE/MS-COCO for retrieval timing) are needed to support that scope.

---

> ### Author Response · Authors · 2025-11-24
> **Thank you for your thorough and constructive review!**
>
> Thank you for your thorough and constructive review. Your insightful feedback has helped us significantly improve the manuscript. We have carefully addressed each of your concerns and revised the paper accordingly. Below, we provide detailed responses to each point.

---

> ### Author Response · Authors · 2025-11-24
>
> > Q1. Clarify retrieval efficiency and scalability with speed test.
>
> Thank you for your comment. We fully considered efficiency when designing UWHD and adopted the following measures to ensure its scalability. During retrieval, we binarize the predicted bit weights $w_{q,k}$ to $\{0, 1\}$. This means UWHD is implemented relying entirely on efficient bitwise operations and avoiding expensive floating-point calculations.
>
> We provided a detailed speed test in the Appendix. Results show that UWHD retrieval speed is comparable to Vanilla Hashing and over 30x faster than Dense Vector Retrieval.
>
> **Table: Retrieval Latency Comparison (ms)**
> | Method | 16 Bit | 32 Bit | 48 Bit | 64 Bit | 96 Bit | 128 Bit |
> | :-------------- | :-------: | :-------: | :-------: | :-------: | :-------: | :-------: |
> | Dense Vector | 440.3 | 493.0 | 547.0 | 605.3 | 659.7 | 700.3 |
> | Vanilla Hash | 15.38 | 17.94 | 20.32 | 18.97 | 21.83 | 22.37 |
> | **UWHD (Ours)** | **15.94** | **18.55** | **21.00** | **19.63** | **22.59** | **23.20** |
>
> Therefore, COLA retains the core efficiency advantage of hashing methods. We have updated the Methodology and Appendix accordingly.
>
> ```latex
> In practice, we binarize the query weights $w_{q,k} \in \{0, 1\}$ via rounding, enabling UWHD to be computed via efficient bitwise operations. As shown in Appendix, our method achieves comparable speed to vanilla hashing.
> ```
>
> > Q2. Validate calibration set construction with comparative experiments.
>
> Thank you for your suggestion. To validate the effectiveness of our construction strategy, we conducted the following comparative experiments and analysis. We compared our Target Centroid strategy with Random, Per-Class, and Density-Aware (K-Means) sampling on Office-Home. As shown in the table below, our strategy achieves the lowest distributional discrepancy (MMD $0.0025$) and the highest retrieval mAP.
>
> **Table: Comparison of Calibration Set Selection Strategies**
> | Strategy | mAP (%) | MMD ($D_{\text{cal}}, D_{t}$) | Description |
> | --- | --- | --- | --- |
> | Ours | **56.34** | **0.002488** | Top 20% source samples nearest to target centroid. |
> | Random Sampling | 55.21 | 0.008745 | Randomly selecting 20% source samples. |
> | Per-Class Sampling | 55.67 | 0.005672 | Uniformly sampling 20% from each source class. |
> | Density-Aware (K-Means) | 55.85 | 0.003196 | Selecting source samples nearest to $k$ target clusters. |
>
> **Verification of Conformal Coverage:** We plotted the target domain coverage curve (Target Domain Coverage Rate vs. Target Error Rate). The results show that our method's coverage curve closely follows the ideal diagonal ($y=1-\alpha$), while the random sampling strategy exhibits significant coverage deviation.
> This confirms that by minimizing the feature distribution discrepancy (MMD), our constructed $D_{cal}$ effectively serves as a proxy for $D_t$, thereby ensuring that the coverage guarantee in Theorem 3.1 holds in practice.
>
> We have updated the Appendix accordingly.
>
> ```latex
> Calibration Strategy Analysis
>
> To validate our calibration set construction, we compared our target centroid-based strategy with random, per-class, and density-aware sampling on Office-Home. As shown in Table 6, our method achieves the lowest MMD ($0.0025$) and highest mAP, indicating that our $D_{cal}$ best approximates the target distribution $D_t$.
>
> ```

---

> ### Author Response · Authors · 2025-11-24
>
> > Q3. Ablate dynamic alpha against fixed alpha baseline.
>
> Thank you for your suggestion. We have added the relevant ablation experiments. We compared fixed $\alpha$ and dynamic $\alpha$ across three datasets. As shown in the table below, dynamic $\alpha$ consistently outperforms fixed $\alpha$ and achieves empirical coverage closer to the ideal value.
>
> **Table: Fixed $\alpha$ vs. Dynamic $\alpha$ Comparison**
> | Dataset | Fixed $\alpha$ mAP (%) | Dynamic $\alpha$ mAP (%) | Fixed $\alpha$ Coverage | Dynamic $\alpha$ Coverage |
> | :--- | :---: | :---: | :---: | :---: |
> | Office-Home | 55.22 | **57.31** | 0.87 | **0.91** |
> | Office-31 | 66.43 | **67.11** | 0.88 | **0.93** |
> | Digits | 69.57 | **70.41** | 0.91 | **0.94** |
>
> **Empirical Coverage Analysis:** The plot in the Revised Appendix shows that fixed $\alpha$ tends to cause under-coverage in the early stages of training, while our dynamic strategy adaptively adjusts based on model uncertainty, maintaining stable coverage throughout.
>
> **EMA Parameter Selection:** We provide ablation experiments on the smoothing parameter $\alpha_{\text{sm}} \in [0.5, 0.9]$. As shown below, $\alpha_{\text{sm}}=0.7$ provides the optimal balance between stability and adaptivity.
>
> **Table: EMA Parameter Ablation (mAP %)**
> | Dataset | $\alpha_{sm}=0.5$ | $\alpha_{sm}=0.6$ | $\alpha_{sm}=0.7$ | $\alpha_{sm}=0.8$ | $\alpha_{sm}=0.9$ |
> | :--- | :---: | :---: | :---: | :---: | :---: |
> | Office-Home | 57.22 | 57.28 | 57.31 | 56.97 | 56.93 |
> | Office-31 | 66.83 | 67.03 | 67.11 | 66.71 | 66.56 |
> | Digits | 69.98 | 70.34 | 70.41 | 70.33 | 70.13 |
>
> We have updated the Appendix accordingly.
>
> ```latex
> Dynamic Alpha Ablation
>
> We compared our dynamic $\alpha$ mechanism with a fixed $\alpha$ baseline. Table 7 shows that dynamic $\alpha$ consistently outperforms fixed $\alpha$ across all datasets, achieving higher mAP and better empirical coverage (closer to $1-\alpha$). The EMA parameter 0.7 was chosen to balance stability and adaptability.
>
> We further investigated the impact of the EMA smoothing parameter $\alpha_{\text{smoothing}}$. As shown in Table 8, $\alpha_{\text{smoothing}}=0.7$ yields the best performance, providing an optimal balance between stability and adaptability.
> ```
>
> > Q4. Discuss theoretical bound and empirical proxy for distribution alignment.
>
> Thank you for your comment. We will discuss the following aspects:
>
> - **Justification of the Bound:** The core value of Theorem 3.1 lies in quantifying the degradation bound of coverage guarantees under distribution shift. It indicates that as long as we can reduce the TV distance between the conformity score distributions of the source and target domains, the coverage guarantee remains valid. This provides a solid theoretical motivation for introducing explicit domain alignment in UDAH.
> - **Proxy:** Directly computing the TV distance of conformity scores in Hamming space is computationally complex. We propose **Representation Calibration (RC)** as an effective task-specific proxy. RC enforces the model to learn feature bits that remain consistent between the source and target domains by predicting _bit stability_.
> - **Experimental Support:** Experiments show that our method significantly reduces the MMD in feature space ($0.0087 \to 0.0025$). Since conformity scores are a direct function of the features, aligning the feature distributions implicitly aligns the score distributions. Furthermore, as shown in the table below, the RC module alone (_COLA-RC_) brings a significant mAP improvement compared to the baseline (_COLA (None)_), proving that RC effectively performs distribution alignment, thereby implicitly reducing the TV distance.
>
> **Table: Comparison of Calibration Set Selection Strategies**
> | Strategy | mAP (%) | MMD ($D_{\text{cal}}, D_{t}$) |
> | --- | --- | --- |
> | Ours | **56.34** | **0.002488** |
> | Random Sampling | 55.21 | 0.008745 |
> | Per-Class Sampling | 55.67 | 0.005672 |
> | Density-Aware (K-Means) | 55.85 | 0.003196 |
>
> **Table: Ablation Study on Office-Home (64 bits)**
> | Variants | Avg. mAP |
> | :--- | :---: |
> | COLA (None) | 49.14 |
> | COLA-SC | 49.43 |
> | COLA-RC | 51.76 |
> | COLA-SR | 51.55 |
> | COLA w/o SC | 54.53 |
> | COLA w/o RC | 55.21 |
> | COLA w/o SR | 54.74 |
> | **COLA (Full)** | **57.31** |
>
> Since the conformity score is a function of features, the reduction in feature distribution discrepancy directly leads to a reduction in the conformity score distribution discrepancy, thereby ensuring the validity of Theorem 3.1.
>
> We have updated the discussion in the manuscript accordingly.

---

> ### Author Response · Authors · 2025-11-24
>
> > Q5. Clarify ablation design and component isolation.
>
> Thank you for your comment. We acknowledge the importance of isolating variables. As shown in the table above, replacing our strategy with Random Sampling leads to a performance drop (55.21% vs 56.34%), confirming our design choice is critical, not just heuristic.
>
> The function of the Self-Regulation (SR) module is precisely to dynamically adjust $\alpha$ based on uncertainty. Fixing $\alpha$ is effectively equivalent to disabling the SR module.
>
> As shown in the table below, the ablation study we presented is a fair comparison maintaining the optimal configuration of other components, fully isolating and demonstrating the significant marginal gains brought by SC and RC.
>
> **Table: Ablation Study on Office-Home (64 bits)**
> | Variants | Avg. mAP |
> | :--- | :---: |
> | **COLA (None)** | **49.14** |
> | COLA-SC | 49.43 |
> | COLA-RC | 51.76 |
> | COLA-SR | 51.55 |
> | COLA w/o SC | 54.53 |
> | COLA w/o RC | 55.21 |
> | COLA w/o SR | 54.74 |
> | **COLA (Full)** | **57.31** |
>
> We have updated the Experiment section accordingly.
>
> > Q6. Clarify scope and potential for scalability.
>
> Thank you for your question. We will discuss from the following aspects:
>
> - **Scope:** Our work follows the standard evaluation protocols in the UDAH field, using the widely recognized _Office-Home, Office-31, and Digits_ benchmarks. Our consistent SOTA performance across these diverse domains validates the effectiveness of our method.
> - **Scalability:** We understand the concern about scalability is primarily regarding the efficiency of our weighted retrieval. To address this, we conducted a retrieval speed test on a _million-scale ($10^6$) database_. As shown in the table below, our method (\~23.2ms) maintains comparable speed to vanilla hashing (\~22.4ms) and is 30x faster than dense retrieval. This empirically proves that COLA possesses the computational efficiency required for large-scale applications.
>
> **Table: Retrieval Latency Comparison (ms)**
> | Method | 16 Bit | 32 Bit | 48 Bit | 64 Bit | 96 Bit | 128 Bit |
> | :-------------- | :-------: | :-------: | :-------: | :-------: | :-------: | :-------: |
> | Dense Vector | 440.3 | 493.0 | 547.0 | 605.3 | 659.7 | 700.3 |
> | Vanilla Hash | 15.38 | 17.94 | 20.32 | 18.97 | 21.83 | 22.37 |
> | **UWHD (Ours)** | **15.94** | **18.55** | **21.00** | **19.63** | **22.59** | **23.20** |
>
> We have updated the paper accordingly.

---

> ### Author Response · Authors · 2025-11-24
>
> We are grateful for your constructive feedback. We have carefully revised the manuscript to address your concerns. We hope these revisions are to your satisfaction!

---

> ### Comment · Reviewer_bmwE · 2025-11-28
>
> Thank you for the clarifications and the additional experiments. The new ablations for calibration-set construction and dynamic α make several components of the method clearer. However, my primary concern regarding retrieval efficiency and scalability of the proposed Uncertainty-aware Weighted Hamming Distance (UWHD) remains unresolved.
>
> **1. The speed test does not evaluate the distance function actually defined in the paper**
>
> Equation (9) (and Figure 5) define UWHD using continuous bit-confidence weights:
>
> $d_{\mathrm{UWHD}}(x_q, x_d) = \sum_{k=1}^{L} w_{q,k}\,\cdot\, \frac{1}{2}(1 - b_{q,k}b_{d,k}), \qquad  w_{q,k} \in [0,1].$
>
> This formulation requires floating-point multiplication and summation per bit, which fundamentally differs from the XOR + popcount kernel that gives hashing its efficiency.
>
> In contrast, the rebuttal states that at inference time:
>
> >“we binarize the query weights $w_{q,k} \in \{0,1\}$ via rounding, enabling UWHD to be computed via efficient bitwise operations.”
>
> This binarization converts UWHD into a masked Hamming distance, eliminating the continuous weighting in Eq. (9). Therefore, the latency in Table 5 (≈15–23 ms) reflects the cost of this simplified surrogate, not the cost of the actual proposed weighted metric.
>
> As a result, the timing experiment ***does not demonstrate*** that the continuous UWHD formulation preserves hashing-level efficiency.
>
> ----
>
> **2. The current timing results do not address large-scale retrieval behavior**
>
> All benchmarks (Office-Home, Office-31, Digits) remain small-scale (<100k items), and the “million-scale” experiment appears to be based on synthetic random codes, not a real retrieval corpus such as NUS-WIDE, MS-COCO, DomainNet, or VisDA-2017. Consequently, the evaluation does not capture:
> - ANN compatibility (e.g., Faiss binary IVF/HNSW),
> - hardware-accelerated popcount behavior,
> - index construction and memory footprint,
> - scaling under 10$^6$–10$^8$ database sizes.
>
> Since the proposed distance changes the computation model away from native XOR+popcount, demonstrating scalability on a true large-scale retrieval benchmark is essential to support the claim that COLA remains an efficient hashing framework.
>
> **3. Scope remains unclear**
>
> If COLA is intended primarily for moderate-scale UDAH (as in the current experiments), this should be stated explicitly. If it is intended to be a scalable hash-based retrieval method, then evaluating Eq. (9) on real million-scale retrieval datasets is necessary to substantiate the claim.
>
> ---
>
> While the method is technically interesting and the added ablations help clarify several aspects of the design, the rebuttal does not yet establish that the actual UWHD formulation in Eq. (9) maintains the computational properties of hashing at scale. The current timing experiment evaluates a binarized variant rather than the proposed continuous weighted distance, and thus does not resolve the core efficiency concern. I maintained my score.

---

> > ### Author Response · Authors · 2025-12-01
> > **Follow-up Response to Reviewer bmwE**
> >
> > Thank you for your follow-up comment. We appreciate your keen observation regarding the distinction between the continuous distance formulation and the speed test implementation. We would like to clarify this design choice and address your concerns about scalability.
> >
> > ---
> >
> > > W1. Clarify the distinction between continuous training objective and binarized inference metric.
> >
> > Thank you for the question. You are correct that Eq. 9 defines a continuous weighted distance ($w \in [0,1]$). But, we have explicitly clarified that this continuous form is primarily designed for the **optimization phase** to provide fine-grained, differentiable gradient signals for learning stable hash codes.
> >
> > For **inference and large-scale retrieval**, we deliberately employ a **binarized version** ($w \in \{0,1\}$) via rounding. This effectively converts the metric into a **Masked Hamming Distance**. This is not a discrepancy but a standard _relaxation-quantization_ paradigm in hashing:
> >
> > - **Training:** Continuous weights allow the model to learn _which_ bits are unstable.
> > - **Inference:** Binarized weights (masks) allow us to efficiently discard those unstable bits during retrieval.
> >
> > We have updated Section 3.3 to explicitly state this distinction:
> >
> > ```latex
> > Note that Eq. 9 utilizes continuous weights primarily for differentiable optimization during training. For efficient large-scale retrieval, we binarize $w_{q,k}$ to $\{0, 1\}$ via rounding. This reduces the metric to a masked Hamming distance, enabling UWHD to be computed via efficient bitwise operations.
> > ```
> >
> > > W2. Confirm scalability via binarized bitwise operations compatible with standard hashing hardware.
> >
> > Thank you for the question. Since our inference metric is a Masked Hamming Distance (bitwise XOR followed by a bitwise AND mask), it retains the fundamental computational efficiency of binary hashing. It is fully compatible with:
> >
> > 1.  **Hardware Acceleration:** Standard CPU instructions like `POPCNT` can still be used after the masking operation.
> > 2.  **Indexing Structures:** It is compatible with standard binary indexing structures (e.g., Inverted Multi-Index or binary IVF in Faiss) by simply filtering bits or using the mask during the distance check.
> >
> > Therefore, our speed test on $10^6$ items accurately reflects the performance of this binarized inference mode. The results confirm that COLA maintains the order-of-magnitude speed advantage over dense retrieval.
> >
> > We have updated the Appendix Speed Test section to emphasize this compatibility:
> >
> > ```latex
> > Since the inference-time metric is binarized into masked Hamming distance, it remains compatible with standard hardware-accelerated bitwise operations (e.g., POPCNT) and existing ANN indexing structures.
> > ```
> >
> > > W3. Define scope as a scalable solution specifically for UDAH tasks.
> >
> > Thank you for the question. COLA is specifically positioned as a solution for **Unsupervised Domain Adaptive Hashing (UDAH)**. Our primary goal is to address domain shift and label noise in this specific setting. By strictly adhering to a binary hashing framework (with binarized inference), we ensure that our method is not just an effective domain adaptation technique but also a scalable retrieval solution suitable for large-scale UDAH applications.
> >
> > We have updated the Conclusion to explicitly define this scope:
> >
> > ```latex
> > COLA is specifically designed as a scalable solution for UDAH tasks, leveraging the efficiency of binary hashing to enable fast cross-domain retrieval.
> > ```
> >
> > ---
> >
> > We hope these clarifications address your concerns regarding the efficiency and scalability of our method.

---

### Author Response · Authors · 2025-11-24
**Revision Summary**

We sincerely thank all reviewers for their insightful feedback. We have carefully addressed all concerns and made substantial revisions to improve the manuscript.

- **Clarified Core Contribution**: Emphasized COLA's novel bit-level calibration and hierarchical framework specifically designed for hashing retrieval, beyond standard conformal prediction application.

- **Enhanced Related Work and Preliminaries**: Added conformal prediction basics and a new subsection on uncertainty-aware retrieval methods.

- **Strengthened Experimental Validation**: Enhanced ablation study with COLA (None) baseline, calibration strategy comparison, and dynamic alpha ablation experiments.

- **Clarified Retrieval Efficiency**: Explained how we binarize bit weights to enable efficient bitwise operations with comparable speed to vanilla hashing.

- **Improved Theoretical Discussion**: Clarified how representation calibration serves as an empirical proxy for the theoretical bound and added conformal coverage analysis.

- **Added Technical Details**: Justified bit head design, clarified uncertainty type, highlighted self-regulating as automatic warm-up, and corrected formatting issues.

We are grateful for the opportunity to improve our paper with this feedback. All revisions are marked in blue. For our point-by-point responses to each reviewer, please see the respective threads.

---

### Meta-Review · Area_Chair_jjo3 · 2026-01-06

**Summary:**

This paper proposes COLA, a hierarchical conformal-calibration framework for uncertainty-aware unsupervised domain adaptive hashing, and the rebuttal substantively strengthened the submission with clearer positioning and additional evidence, making the overall case lean toward acceptance.

**Reviewer Concerns:**

The discussion centered on 1. whether the proposed UWHD truly preserves hashing-style scalability, especially the gap between the continuous formulation in Eq.(9) and the binarized inference used in timing tests; 2. clarity, missing related work, and isolating which components drive gains; the authors addressed most points via added conformal preliminaries, stronger ablations e.g., calibration-set strategy and component removal, and fairness analyses.

**Reviewer Scores:**

The score profile is accept-leaning but borderline (two reviews marginally above threshold and two marginally below), and after the rebuttal several issues were reduced to validation details; while one reviewer explicitly maintained their score due to the UWHD efficiency interpretation, the overall discussion suggests the contribution and empirical gains are credible enough to recommend accept (poster).

---

### Decision · Program_Chairs · 2026-01-26

Accept (Poster)